cognition/neuroscience/psychology

dog, word processing, event-related potentials, phonetic similarity

**Author for correspondence:**
L. Magyari
e-mail: lillamagyari@gmail.com

# Event-related potentials reveal limited readiness to access phonetic details during word processing in dogs

L. Magyari[1,2,3], Zs. Huszár[2,4], A. Turzó[1,2] and A. Andics[1,2]

[1]MTA-ELTE 'Lendület' Neuroethology of Communication Research Group, Hungarian Academy of Sciences, Eötvös Loránd University, Budapest 1117, Hungary
[2]Department of Ethology, Institute of Biology, Eötvös Loránd University, Budapest 1117, Hungary
[3]Department of Cognitive Psychology, Institute of Psychology, Eötvös Loránd University, Budapest 1064, Hungary
[4]Department of Cognitive Science, Faculty of Natural Sciences, Budapest University of Technology and Economics, Budapest 1111, Hungary

LM, 0000-0002-1188-2593

While dogs have remarkable abilities for social cognition and communication, the number of words they learn to recognize typically remains very low. The reason for this limited capacity is still unclear. We hypothesized that despite their human-like auditory abilities for analysing speech sounds, their word processing capacities might be less ready to access phonetic details. To test this, we developed procedures for non-invasive measurement of event-related potentials (ERPs) for language stimuli in awake dogs ($n = 17$). Dogs listened to familiar instruction words and phonetically similar and dissimilar nonsense words. We compared two different artefact cleaning procedures on the same data; they led to similar results. An early (200–300 ms; only after one of the cleaning procedures) and a late (650–800 ms; after both cleaning procedures) difference was present in the ERPs for known versus phonetically dissimilar nonsense words. There were no differences between the ERPs for known versus phonetically similar nonsense words. ERPs of dogs who heard the instructions more often also showed larger differences between instructions and dissimilar nonsense words. The study revealed not only dogs' sensitivity to known words, but also their limited capacity to access phonetic details. Future work should confirm the reported ERP correlates of word processing abilities in dogs.

## 1. Introduction

Certain capacities for auditory processing of language, such as analysing speech sounds and mapping signals into meaning, are

**Table 1.** Stimuli used in the experiment for each condition. Conditions are shown in the first row (capitalized). (Each word (in Hungarian) is followed by their International Phonetic Alphabet phonetic transcription in square brackets. The second column shows the lexical meaning of the instruction words in English (italics). Consonant–vowel structure of words in each row is shown in the last column (C, consonant; V, vowel).)

| WORDS | meaning | SIMILAR | NONSENSE | structure |
|---|---|---|---|---|
| Fekszik [fɛksik] | *lay down* | Fakszik [fɒksik] | Matszer [mɒtsːɛr] | CVCCVC |
| Marad [mɒrɒd] | *come* | Merad [mɛrɒd] | Hefegy [hɛfɛɟ] | CVCVC |
| Gyere [ɟɛrɛ] | *you can go* | Gyare [ɟɒrɛ] | Dime [dimɛ] | CVCV |
| Mehetsz [mɛhɛtsː] | *stay* | Mihetsz [mihɛtsː] | Rekaksz [rɛkɒks] | CVCVCC |

shared abilities across species [1]—ones that are shared not only with humans' closest evolutionary relatives, non-human primates [2] but also with other species, e.g. dogs and parrots [3,4]. Dogs can distinguish human speech sounds [5–7], similarly to other non-primate species (e.g. cat [8] and Japanese quail [9]) and follow tape-recorded instructions in the absence of other gestural information [5,10,11]. A behavioural study using a head-orientation paradigm has shown similar functional hemispheric asymmetries for dogs compared with humans in processing vocalizations [12]. A study from our laboratory has recently also shown similarities in the neural correlates of human and dog word processing [13]. However, differences in learning words and in vocabulary size between dogs and humans are also apparent. Dogs that are responsive to hundreds or a thousand words [3,14] are extremely rare. While there are differences in lexical capacity between dogs and human adults, humans also do not process words in the same way from birth, but word processing capacities go through on developmental changes. We hypothesized that dogs' word processing abilities may be different from capacities of human adults and are more similar to that of young infants who are in an early stage of the development of word processing skills. Here, we aim to investigate this by measuring electrophysiological brain activity in dogs.

An increasing body of evidence concurs that younger infants' word recognition and learning is different from that of older ones. Infants become efficient in processing phonetic details of words, an important prerequisite for developing a large vocabulary, between 14 and 20 months. Younger infants (around 14 months) fail to associate phonetically similar novel words (i.e. non-existing words in English) such as *bih* or *dih* to different objects in word learning situations [15,16]. They also fail to differentiate nonsense words from phonetically similar known words [17]. In an event-related potential (ERP) study of auditory word recognition [17], ERPs did not differ when 14-month-old infants were listening to words they knew and to phonetically similar nonsense words, while ERPs were different for phonetically dissimilar nonsense words. By contrast, 20-month-old infants showed different ERPs for known words compared with both types of nonsense words.

Here, we study the electrophysiological correlates of word processing in dogs, following up directly on the infant ERP study by Mills *et al.* [17]. Our study is among the first ones to apply electroencephalography (EEG) fully non-invasively in awake, untrained, cooperating animals. There is a long tradition of continuous EEG research on dogs applying invasive methods, especially in studies of neurological disorders, like epilepsy (e.g. [18,19]). An earlier EEG study investigating speech perception in tranquilized dogs found an auditory evoked response which discriminated consonant sounds in a categorical manner [7]. Fully non-invasive EEG has been applied in earlier studies on awake but trained dogs [20,21]. To date, only one study measured EEG in awake, untrained dogs but with a minimally invasive method (needles inserted under the skin) [22].

## 2. Results and discussion

Dogs were presented with three different word-types: (i) with familiar instruction words (WORDS condition) (based on owner report; see Material and methods), (ii) with phonetically similar nonsense words (SIMILAR), and (iii) with dissimilar nonsense words (NONSENSE) (table 1). During EEG measurement, dogs heard each type of words 80 times in a random order through loudspeakers.

There is no standard way of data cleaning of awake, non-invasive dog EEG; however, many movement artefacts are present in the raw signal. Here, we conducted two different procedures for data cleaning. First, we applied a multi-level method for artefact rejection (figure 1), which involved

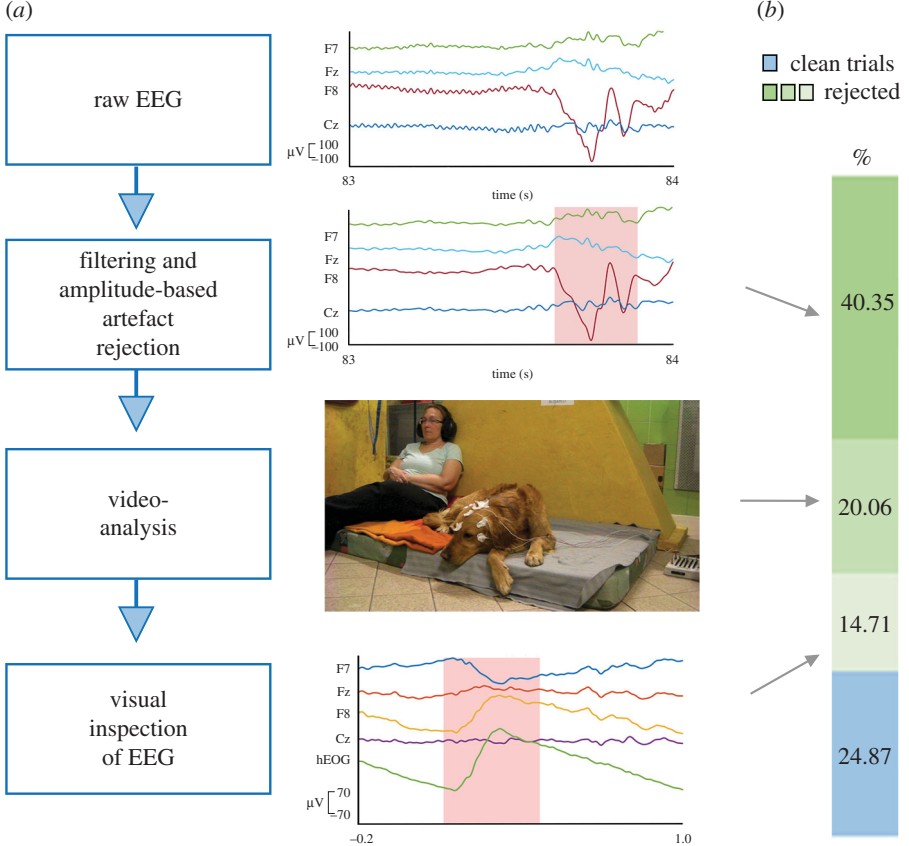

**Figure 1.** Procedure for the multi-level artefact rejection. (*a*) Artefact rejection pipeline. Blue boxes show the consecutive steps of artefact rejection (left), graphs (right) illustrate features of EEG data for rejection (e.g. bottom graph shows EEG pattern typical for eye blink) with time on *x*-axis (s) and electric potentials (μV) and electrodes on *y*-axis, and light red boxes show electric potentials with artefacts; image (right) is a still from the video-recording of the experiment. (*b*) Percentage of rejected and clean trials.

both quantitative (e.g. filtering and amplitude-based trial rejections) and qualitative (i.e. visual inspection of video-recordings and of EEG patterns) steps and which complies with standards of EEG research on human infants [23]. According to our knowledge, combining visual data exclusion with amplitude-based artefact rejection has been applied here for the first time on the EEG of awake dogs. Second, we also cleaned the raw data of the same participants by only applying a quantitative step, filtering and amplitude-based artefact rejection which has been used in other studies (e.g. [21,22]). During the amplitude-based cleaning procedure, we used stricter thresholds for rejecting trials compared with the amplitude-based artefact rejection step of the multi-level cleaning pipeline. We conducted the same statistical analysis on the two cleaned datasets, which showed similar condition differences with partially different timings.

One-way repeated-measures ANOVA showed no differences in the number of clean trials between conditions after either the multi-level ($F_{2,32} = 1.143$, $p$[Huynh-Feldt (HF)] = 0.167) or the amplitude-based cleaning procedure ($F_{2,32} = 0.455$, $p$[HF] = 0.606; see table 2 in Material and methods). In the multi-level data cleaning, 75% and in the amplitude-based procedure, 53% of the trials were rejected.

Following artefact rejection, segments were averaged separately for each condition and participant ($n = 17$) for two electrodes (Fz and Cz) in selected time-windows for each condition. First, we checked whether there was an effect of condition (word-types) on the ERP between 200 and 400 ms which is a time-window for ERP differences in Mills *et al.*'s similar experiment [17] with infants. Second, we conducted a sliding time-window analysis to determine relevant time-windows for further statistical analysis (see Material and methods on details for time-window selection).

Using Bayes factors (BFs) [24], we also tested the lack of condition differences, i.e. differences in the ERP amplitudes of those conditions which were not significantly different by repeated-measures ANOVA in those time-windows where other pairs of conditions showed a difference. We tested the strength of evidence for the null hypothesis, i.e. the strength of evidence for no differences between conditions.

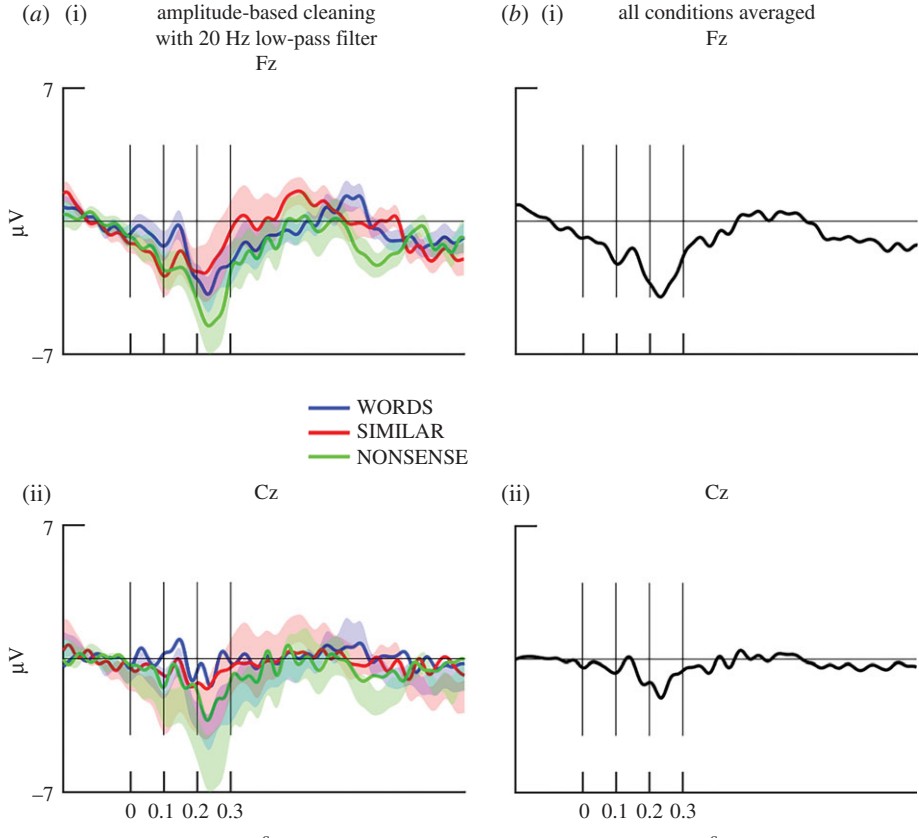

**Figure 2.** Grand-averaged ERPs in three conditions (*a*) and averaged for all trials (*b*) at two electrode sites, Fz (i) and Cz (ii) after an amplitude-based cleaning procedure with a 20 Hz low-pass filter from 200 ms before onset of stimulus presentation to 1 s (*y*-axis). Vertical line shows timing of word onset (0 s), and 100, 200 and 300 ms after word onset. Coloured shades show standard errors.

**Table 2.** Mean, standard deviation, minimum and maximum of the number of clean trials for each condition after the multi-level (left) and the amplitude-based (right) data cleaning procedures.

| | multi-level cleaning | | | | amplitude-based cleaning | | | |
|---|---|---|---|---|---|---|---|---|
| | mean | s.d. | minimum | maximum | mean | s.d. | minimum | maximum |
| WORDS | 18.82 | 11.21 | 10 | 51 | 38 | 12.63 | 23 | 66 |
| SIMILAR | 20.82 | 10.74 | 10 | 47 | 37.94 | 13.94 | 18 | 72 |
| NONSENSE | 19.41 | 10.29 | 10 | 45 | 36.94 | 13.35 | 20 | 70 |

Following Kass & Raftery [25], we interpreted BF between 3.2 and 10 as a substantial evidence for no differences, BF larger than 10 as a strong evidence and BF between 1 and 3.2 as 'not worth more than a bare mention' [25, p. 777].

For better visualization of the general shape of the ERP and an easier detection of, e.g. primary auditory response in the data, we set the low-pass filter to a lower value (20 Hz). Then, the trials were cleaned by the amplitude-based method (see also electronic supplementary material, results and discussion). Segments were averaged separately for each condition, participant and electrodes (Fz, Cz) (figure 2*a*), and segments were averaged across conditions for each electrode (figure 2*b*). The ERPs show a small negative peak around 100 ms and a larger negative peak between 200 and 300 ms at all conditions and at the grand-average across conditions. The smaller negative peak is similar to the N100 component, a primary auditory response which is often observed in humans [26] and also observed by Howell *et al.* in dog EEGs [22]. This strongly suggests that the measured EEG signal is sensitive to perceptual and cognitive processing in the dogs' brain. However, further research should confirm this with an experiment which is designed to modulate the N100 component. Future studies

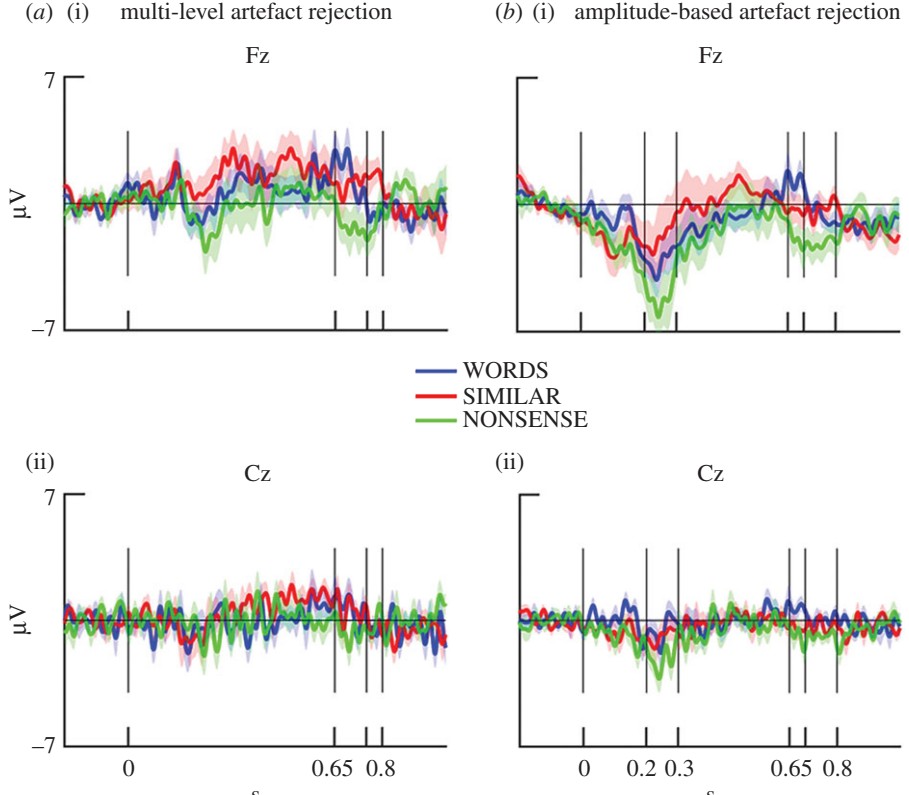

**Figure 3.** Grand-averaged ERPs in three conditions after the multi-level and the amplitude-based cleaning at two electrode sites, Fz (a) and Cz (b) from −200 ms before onset of stimulus presentation to 1 s (y-axis). Vertical line shows timing of word onset (0 s), and the boundaries of condition differences (0.75 on the left and 0.7 on the right x-axis are not labelled). Coloured shades show standard errors.

could also examine whether the negativity around 200–300 ms is stimulus-specific or also reflects a more general auditory response.

Figure 3 shows grand-averaged ERPs for all conditions at electrodes Cz and Fz after both data cleaning procedures (see details described in Material and methods). For the data cleaned with the multi-level method, there was no effect of condition, electrodes or their interaction between 200 and 400 ms after stimulus presentation by a two-way (word-types, electrodes) repeated-measures ANOVA (word-types: $F_{2,32} = 1.131$, $p[HF] = 0.3351$; electrodes: $F_{2,32} = 1.070$, $p = 0.316$, interaction: $F_{1,16} = 1.717$, $p[HF] = 0.1957$). For the data cleaned with the amplitude-based artefact rejection method, there was a trend for both condition- and electrode-differences (word-types: $F_{2,32} = 3.171$, $p[HF] = 0.0554$; electrodes: $F_{1,16} = 3.383$, $p = 0.0845$, interaction: $F_{2,32} = 1.955$, $p[HF] = 0.1604$).

For the data cleaned with multi-level artefact rejection, the sliding time-window analyses resulted in two similar time-windows for certain pairs of conditions. Differences between WORDS and NONSENSE conditions were indicated by a main effect of word-types in the time-window between 650 and 750 ms ($F_{1,16} = 5.65$, $p = 0.03$, partial $\eta^2 = 0.26$). Averaged ERP amplitudes for WORDS were more positive than for the other condition (figure 4). Electrodes and their interaction with word-types had no effect.

Averaged amplitudes of the SIMILAR condition were more positive in the time-window from 650 to 800 ms compared with the NONSENSE condition (figure 4) ($F_{1,16} = 5.66$, $p = 0.03$, partial $\eta^2 = 0.26$). Electrodes and their interaction with word-types had no effect. There was no time-window with significant differences between WORDS and SIMILAR conditions. We also tested the differences of these two conditions in the selected time-windows of the other condition pairs, but there was no statistically significant difference between conditions (650–750 ms: $p = 0.74$, partial $\eta^2 = 0.007$; 650–800 ms: $p = 0.61$, partial $\eta^2 = 0.02$). BFs showed a substantial evidence for no difference between WORDS and SIMILAR in both time-windows (650–750 ms: BF = 3.691 ± 3.71%; 650–800 ms: 3.736 ± 0.78%).

For the data cleaned with amplitude-based artefact rejection, sliding time-window analysis revealed three time-windows for condition differences (figures 3 and 4). ERPs for WORDS ($F_{1,16} = 5.744$, $p = 0.02911$, partial $\eta^2 = 0.26$) and for SIMILAR ($F_{1,16} = 7.979$, $p = 0.0122$, partial $\eta^2 = 0.33$) were more positive compared with NONSENSE between 200 and 300 ms. There was no difference between

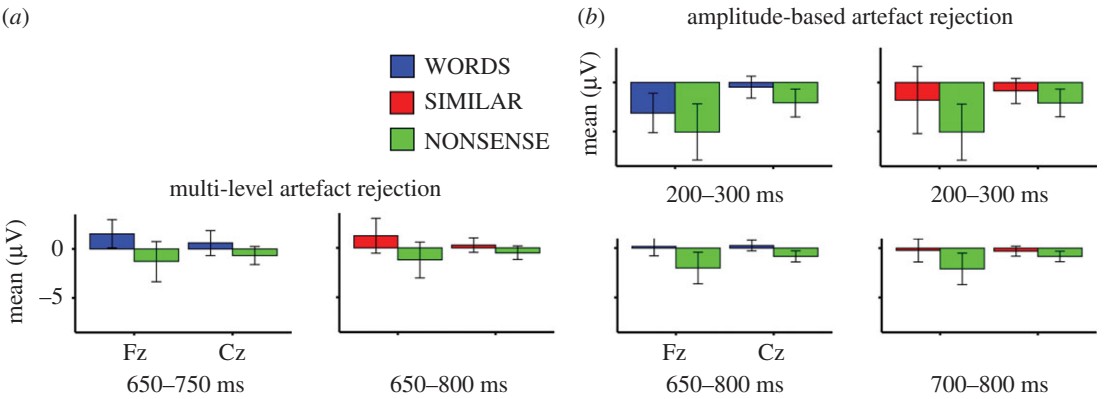

**Figure 4.** Mean ERP (*y*-axis) in the selected time-windows for the data cleaned with multi-level (*a*) and with amplitude-based artefact rejection (*b*) at Fz and Cz electrodes (left and right sides of *x*-axis). Error bars show 95% confidence interval.

WORDS and SIMILAR conditions in this time-window ($F_{1,16} = 0.379$, $p = 0.5468$). BF also showed a substantial evidence for no difference (BF = 3.568 ± 2.14%). When ERPs for WORDS and NONSENSE were compared in this time-window, there was also a main effect of electrodes ($F_{1,16} = 10.948$, $p = 0.0044$). ERPs at Fz were more negative than at Cz. Between 650 and 800 ms, ERP for NONSENSE was more negative than for WORDS ($F_{1,16} = 8.596$, $p = 0.0098$, partial $\eta^2 = 0.349$). Between 700 and 800 ms, ERP was also more negative for NONSENSE than for SIMILAR ($F_{1,16} = 6.354$, $p = 0.0227$, partial $\eta^2 = 0.284$). In this later time-window, there was also an interaction between word-types and electrodes ($F_{1,16} = 5.084$, $p = 0.0385$, partial $\eta^2 = 0.24$). While ERP for NONSENSE was more negative than the ERP for SIMILAR at Fz ($t_{16} = 2.535$, $p = 0.0221$), there was only a trend for difference at Cz ($t_{16} = 1.807$, $p = 0.0899$). There was no significant difference between WORDS and SIMILAR either in the 650–800 ms ($F_{1,16} = 1.106$, $p = 0.309$, partial $\eta^2 = 0.064$) or in the 700–800 ms time-window ($F_{1,16} = 0.031$, $p = 0.862$, partial $\eta^2 = 0.002$). BF showed only a weak evidence (not worth more than a bare mention) for the lack of differences between WORDS and SIMILAR in the 650–800 ms time-window (BF = 1.742 ± 2.56%), and it showed a substantial evidence for no difference between the same conditions in the 700–800 ms time-window (BF = 3.458 ± 9.85%).

The results show similar effects between conditions after the amplitude-based and the multi-level cleaning in the late time-windows. However, the condition differences between 200 and 300 ms were found only after the amplitude-based artefact rejection. Graphs of individual ERPs of each dog are provided in the electronic supplementary results (electronic supplementary material, figure S2).

We also tested whether the magnitude of ERP effect between instruction words and phonetically dissimilar nonsense words was influenced by the dog's experience with the instructions, i.e. the frequency with which owners used them, and by the dog's behavioural consistency in response to those instructions, as reported by the owner. For this, we calculated the difference of average ERP values at Fz between condition pairs in the corresponding time-windows for each dog and the average owner-reported scale values for the frequency of usage and behavioural consistency across the four instruction words (see Material and methods). Then, we split dogs to two groups based on the median of frequency of usage ($n = 8$, mean = 4.59, s.d. = 0.23; $n = 9$, mean = 5, s.d. = 0) and of behavioural consistency ($n = 8$, mean = 4.22, s.d. = 0.28; $n = 9$, mean = 4.86, s.d. = 0.13). For dogs who heard the instruction more often, average ERP amplitude differences between WORDS and NONSENSE conditions at Fz were higher compared with the other group by the Wilcoxon rank-sum test in the time-window between 650–750 ms ($W = 14$, $p = 0.036$, effect size $r = -0.51$, confidence interval (CI) = (−8.35, −0.33)) and between 650–800 ms ($W = 14$, $p = 0.036$, effect size $r = -0.51$, CI = (−6.01, −0.03)) for the data cleaned with multi-level and with amplitude-based artefact rejection, respectively (figure 5). There was no difference between ERP amplitudes of the two groups in the early (200–300 ms) time-window of the data cleaned with amplitude-based artefact rejection ($W = 34$, $p = 0.888$). There was also no significant difference in the ERPs between the groups of different behavioural consistency, and there was no difference between average ERP differences for WORDS and SIMILAR conditions and for SIMILAR and NONSENSE conditions either after amplitude-based or multi-level cleaning.

In the data cleaned with amplitude-based artefact rejection, we checked at electrode Fz whether the same dogs or different ones show an early and a late effect between WORDS and NONSENSE. In both time-windows, there were 12 dogs for which the direction of the difference between these two conditions

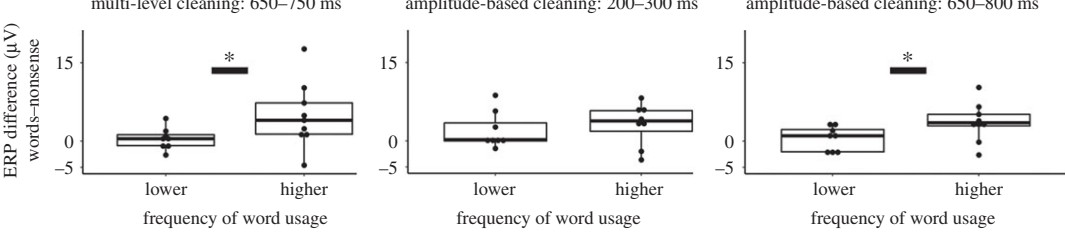

**Figure 5.** The effect of dog owners' word usage frequency on the average ERP difference of WORDS and NONSENSE conditions at Fz in three time-windows. Two groups of dogs based on the owner's average usage frequency of the instruction words (split by median) are shown on the x-axis, and ERP amplitude differences are shown on the y-axis. Asterisks (*) represent significant differences between groups.

matched the direction of the difference of the grand-averages. Ten of these 12 dogs are the same across the two time-windows. Therefore, approximately the same dogs show similar condition differences in both time-windows.

This study provides the first electrophysiological evidence, to our knowledge, for word processing in dogs, revealing its temporal dynamics. Dogs' ERP responses did not differentiate instruction words (WORDS) and phonetically similar nonsense words (SIMILAR), but ERP for both WORDS and SIMILAR was different from ERP for phonetically dissimilar nonsense words (NONSENSE). This suggests that dogs process instruction words differently from dissimilar nonsense words.

ERP differences between WORDS and NONSENSE conditions appeared between 650 and 800 ms following word-onset, towards the end of the words ($M_{\mathrm{duration}} = 650$ ms) in a dataset cleaned by multi-level methods, and there was also a difference between 200 and 300 ms when data were cleaned only by an amplitude-based procedure. In comparison, word-familiarity effect in human infants is observable approximately between 200 and 500 ms following word-onset [17,27], and it is considered to reflect the mental representation of word forms (e.g. [28,29]). In human adults, differences between pseudowords and known words elicited ERP differences as early as 100 ms following word-onset in a picture-word priming paradigm [28]. In the current study, ERP differences could already be observed around the same time relative to stimulus presentation as the timing of ERP differences typically reported in human infant and adult studies. This suggests that dogs can discriminate words as humans do.

However, the results are based on the EEG measurement of 17 dogs with a low number of trials following the multi-level artefact rejection analysis. Further studies are needed to replicate these findings with more robust effects. Similar effects were also present after the amplitude-based cleaning procedure which involved more trials in the subject-averages. The amplitude-based cleaning procedure was conducted on the same data; therefore, the results based on this procedure do not confirm the results of the data cleaned with multi-level artefact rejection, and nevertheless, it provides a methodological insight. The amplitude-based artefact cleaning procedure shows that the results are not too sensitive to the method for cleaning, because similar condition differences were found as after the multi-level cleaning procedure. However, the extra trials of the amplitude-based artefact cleaning were rejected during the multi-level artefact rejection procedure. Therefore, these trials might contain artefacts. But it is important to note that the cleaning procedure of the multi-level procedure was very conservative. For example, trials with a correlation between electrooculography (EOG) channels (F7, F8) and the Fz–Cz channels were also rejected. This was motivated by the assumption that any artefact from eye-movements showing up on the Fz and Cz channels will be also present on F7 and F8 because those channels are more sensitive for eye-movements. But task-dependent, cognitive effects can also show up on all channels, including F7 and F8. Hence, we think it is also likely that the multi-level data cleaning procedure was too conservative and clean trials without any artefacts also got rejected. Similar effects could show up after the amplitude-based cleaning when more trials were included because we eliminated some of the trials containing the experimental effect in the multi-level procedure. These assumptions should be confirmed in future studies.

Future research should also confirm whether word processing in dogs has similar timing as in humans because our results showed a similar temporal pattern (i.e. an effect starting from 200 ms) only in the analyses of the data cleaned with the amplitude-based artefact rejection. Visual inspection of the ERPs also showed differences between conditions around 200–300 ms at Fz in the data cleaned with multi-level artefact rejection, but these differences did not reach significance (figure 3a(i)), perhaps because of the low number of clean trials following rigorous artefact rejection.

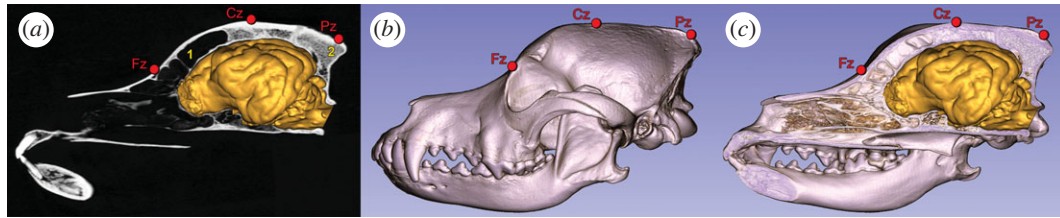

**Figure 6.** Placement of three electrodes, Fz, Cz and Pz, relative to the skull and the brain of a border collie (as an example of a dog with mesocephalic head type). (*a*) Computerized tomography (CT) image in the midsagittal plane of the skull with the brain. (1) Frontal sinus and (2) external occipital protuberance. (*b*) Volume-rendered CT scan of the skull, left lateral view. (*c*) The volume-rendered CT scan of the skull, which was cut in the midsagittal plane, together with the brain.

The ERP effects in the late time-windows were found 650–800 ms after stimulus onset, while the duration of the auditory stimuli was, on average, 650 ms. This raises the question of whether the obtained differences are owing to stimulus offset caused by condition-dependent muscle movements (i.e. shifting of position and tension of the ears). We think that given the rigorous method for artefact rejection in the multi-level cleaning, such a confound has a low possibility. Moreover, even if ERP effects were because of muscle movements, those would still show that dogs differentiate word-categories. Hence, such a confound would not undermine the claim about dogs' ability for discriminating known words from nonsense words; however, it would not provide electrophysiological evidences for word processing in dogs which was the primary aim of this study.

Regarding the spatial distribution of the ERP differences, figures 3 and 4 show that condition differences at Fz are larger than at Cz. This may show that ERP correlates of word processing are more focused above frontal compared with central brain areas; however, the attenuation at Cz can also reflect the spatial proximity of this electrode to the reference (Pz) (see Electrophysiological recording; figure 6).

Although the same speech sounds build up the words in all conditions, condition differences in the ERP could have also occurred if there were systematic differences in the initial speech sounds of the words across the different conditions. In this case, the ERP effects could reflect speech sound processing differences, regardless of whether the words were meaningful or nonsense. The initial consonants of words in the NONSENSE condition are indeed different from the initial consonants of the words in the WORDS and SIMILAR conditions. Nevertheless, as there were no systematic differences between the initial consonants of the conditions in either mode of articulation (cf. [17], see Stimuli) or in frequency as a word-starting phoneme (see the electronic supplementary material, table S1 in methods), it is improbable that the difference in initial consonants explains the ERP differences between conditions. Therefore, it is likely that the reported ERP effects reflect word familiarity processing rather than speech sound processing.

The positive association between word usage frequency by the dog's owner and individual ERP effects further supports the word familiarity account. Word processing in humans is also sensitive to word frequency [30,31]. This association was only found in the late time-window of the ERP effect of the WORDS – NONSENSE condition difference. Therefore, it remains an open question whether the two time-windows reflect different aspects of word processing.

The overlapping time-windows of the ERP differences between WORDS and NONSENSE and between SIMILAR and NONSENSE suggest that dogs process words and phonetically similar nonsense words in a similar way. Previous research on dogs' speech sound discrimination abilities suggests that this ERP insensitivity to phonetic details is not owing to perceptual or representational incapability. Fukuzawa *et al.* [5] showed in a behavioural experiment that dogs listening to instruction words perceive differences between initial consonants as well as between vowels in the first syllable. Dogs listened to commands (sit and come) and their modified versions in which phonemes were changed. Dogs noted the difference in alternation of both the first consonants (e.g. [tʃɪt] instead of sit) and the vowels (e.g. [sæt] instead of sit), as shown by the decline of responses to the alternated commands. Based on research on human infants, we have also no reason to assume that the vowel manipulation we applied here (swapping [i], [ɒ] and [ɛ] across conditions in the first vowel position) would be perceptually less salient than the consonant manipulation applied by Mills *et al.* [17]. In human development, infants become sensitive earlier to vowel changes than to consonant changes. For example, Mills *et al.*'s study [17] found that 14-month-olds did not differentiate words with changed initial consonants, while another study with 14-month-olds [32] revealed ERP sensitivity to

vowel mispronunciations. The presence of vowel-harmony in Hungarian is one more reason to pay more attention to vowels for individuals living in a Hungarian language environment, like our dogs are. Indeed, infants learning Hungarian perceive vowel-harmony of words already at 13 months in contrast with 13-month-old French infants [33]. What may then cause the reported insensitivity in dog ERPs to phonetic details, if it is not perceptual or representational incapability? It has been proposed that younger infants might not process the phonetic details of words in certain experimental situations because of attentional and processing demands or weaker representations owing to less familiarity with words [17,34,35]. Specifically, even though 14-month-old human infants may not differentiate words from phonetically similar nonsense words in certain experiments [17], they showed phonetic differentiation effects, for example, in word learning situations wherein novel names should be associated with already familiar objects (e.g. [34]). Therefore, we speculate that the similarity of dogs' ERPs for instruction words and phonetically similar nonsense words reflects attentional and processing biases, rather than perceptual constraints, similarly to the case of human infants. However, future research needs to confirm this assumption.

We developed a procedure to study word processing with fully non-invasive EEG in untrained companion dogs. The results suggest that companion dogs are able to discriminate words they often heard from nonsense words. Individual differences in the ERP effects may indicate that dogs' word processing capacities are sensitive to the frequency with which dogs encounter the words. This is in line with earlier studies which showed analogies of word processing between dogs and humans [12,13]; however, the present ERP findings should be confirmed by future studies with more participants and trials. Based on the lack of differences between the ERPs for known and for phonetically similar nonsense words, we propose that dogs may have a limited word processing capacity. We suggest that this limitation is not owing to insufficient perceptual discrimination, but it might reflect different attentional allocation compared with human adults. Future research should confirm whether the reduced readiness to process phonetic details may be one reason that incapacitates dogs from acquiring a sizeable vocabulary.

# 3. Material and methods

## 3.1. Participants

Healthy companion dogs' EEG data ($n = 17$; six females; nine mix breeds and 11 from six different breeds), ranging from 1.5 to 9.5 years of age ($M = 5.44$, s.d. $= 2.68$), were analysed. We recruited and selected 44 dogs in Hungary on the basis of an online language-questionnaire completed by owners. Owners and their dogs were invited to participate in the current study if owners' indicated on a 5-point Likert scale that they use four Hungarian instructions (table 1) many times a week and their dogs often show a behaviour consistent with the instructions. From our analysis, we excluded data from one dog whose head-length and wither height was more than two s.d. above participants' average, 14 dogs who precluded electrode placement and 12 dogs because of high numbers of movement artefacts during the three-step data cleaning procedure (see later). The ratio of dogs invited for the study and dogs whose data were included in the data analysis (38%) is similar to the inclusion rate of ERP studies with human infants (e.g. [36]).

## 3.2. Stimulus material

Based on the questionnaire filled by the owners, we selected those four, two-syllable-long instructions which were most frequently used by Hungarian-speaking dog owners (table 1). We also created phonetically similar and phonetically dissimilar nonsense variants of each instruction. Phonetically similar versions were created by changing the first vowel of the instruction word to the first vowel of one of the other instructions, as earlier studies [5,6] showed evidence of dog's ability for discrimination of vowels (see the vowel space of the interchanged vowels in the electronic supplementary material, figure S1). The dissimilar nonsense words were created by mixing the sounds of all four instructions (table 1). In Hungarian, word-stress is fixed and comes on the first syllable of words, so stress pattern was the same for all stimuli. The frequency of words starting with the initial consonants of all stimuli is shown in the electronic supplementary material, table S1. Initial consonants of words were the same in the WORDS and SIMILAR conditions and different from the words in the NONSENSE condition. There were three words with voiced and one word with voiceless initial consonants in each condition.

The mode of articulation of the initial consonants was not fully balanced, but differences were not systematic. When a certain type of consonants was present in one condition while missing from the other, this type of consonant occurred not more than once in that condition. There were one stop consonants and one fricative in each condition, one flap and one nasal in the NONSENSE condition, and two nasals in the other two conditions.

Each word was spoken by two female Hungarian native speakers in a friendly tone of voice and neutral intonation, digitized at forty-four 100 Hz and recorded twice. A total of 48 sounds (four (words) × three (conditions) × two (speakers) × two (versions)) were used as stimuli. Average loudness (root mean square values) was equated across stimuli. The duration of sound-files ($M = 649.89$ ms, s.d. = 160, $F_2 = 0.41$, $p = 0.66$), or average pitch-level as measured by the PRAAT software [37] ($M = 221.27$, s.d. = 10.33, $F_2 = 2.11$, $p = 0.13$), did not differ across conditions in one-way ANOVA analysis. Each stimulus was played five times in a semi-randomized order during the experiment. We created four experimental lists. Each list contained a different semi-randomized order of the 240 stimuli (80 trials in each condition). These four lists were created with the restriction that stimuli from the same speaker or from the same condition did not follow each other more than three times. Dogs were assigned randomly to one of the experimental lists.

## 3.3. Procedure

EEG recording took place in the laboratory of the Ethology Department of the Eötvös Loránd University in Budapest. The room was not soundproof but, importantly, the only ambient environmental noise was continuous and soft (air conditioner). First, participants were allowed to freely explore the room. When the dog was ready to settle down, we asked the owner to sit on a mat placed on the floor and to call her dog to settle next to her. During the experiment, owners were asked to stay sitting next to their dog as if it would be a relaxation period for them and for their dog. Dogs were relaxing most of the time by lying down, and in some rare cases by sitting. When a dog stood up during the experiment, the owner was asked to try to put the dog back to lying position. If the dog wanted to leave the mat, the experiment was aborted. Two experimenters applied electrodes on the dog's head. The owner and the two experimenters were present in the room during the EEG recording, but all of them were out of the view of the dog. Auditory stimuli were played from two loudspeakers placed on the ground in the right and left front of the dog at 1.5 m distance. Loudness was calibrated to $65 \pm 10$ dB. The auditory stimuli and the triggers to the EEG amplifiers were presented by PSYCHOTOOLBOX [38] in MATLAB 2014b with 2500–3000 ms stimulus onset asynchrony. During EEG recording, owners were wearing sound-proof headphones, owners and dogs were not facing each other and dogs were video-recorded. The experiment was approximately 11 min long after applying the electrodes.

## 3.4. Electrophysiological recording

Electrode placement followed a canine EEG set-up developed and validated in our laboratory for sleep studies [39], which has been used by several sleep studies (e.g. [40,41]). Surface-attached scalp electrodes (gold-coated Ag | AgCl) were fixed with EC2 Grass Electrode Cream (Grass Technologies, West Warwick, RI, USA) on the dogs' head and placed over the anteroposterior midline of the skull (Fz, Cz, Pz), and on the zygomatic arch (os zygomaticum), next to the left and right eyes (EOG) (figure 6). The ground electrode was placed on the left musculus temporalis. Fz, Cz and EOG derivations were referred to Pz [40]. The reference electrode (Pz), Cz and Fz were placed on the dog's head at the anteroposterior midline above the bone to decrease the chance for artefacts from muscle movements. Placement of the reference electrode is similar to the reference applied in Howell and her colleagues' minimally invasive EEG study on dogs [22], in which the reference was placed in the midline of the neck. In our study, Pz was also placed posteriorly to the active electrodes. However, it was placed on a head-surface above the back part of the external sagittal crest (crista sagittalis externa) at the occipital bulge of dogs where the skull is usually the thickest and under which either no brain or only the cerebellum is located depending on the shape of the skull [42] (figure 6). Therefore, this placement provides a good base for reference as less brain activity can be seen there and provides a good control for artefacts as it is close to the other electrodes. On the other hand, Pz as a reference could lead to an attenuated effect on Cz compared with Fz, because it is closer to Cz.

Impedances for the EEG electrodes were kept at least below 15 kΩ, but mostly below 5 kΩ. The EEG was amplified by a 40-channels NuAmps amplifier (Compumedics Neuroscan) with applying DC-recording, and it was digitized at a 1000 Hz sampling rate.

**Table 3.** Time-window selection based on effect of word-types of two-way repeated-measures ANOVA in 100 ms long time-windows. (Only time-windows with a significant effect of word-type as a three-level factor or with a significant interaction between word-types and electrodes are shown (see the first row). Significance of the main effect or interaction of word-types in the pairwise comparison of conditions is shown from the second to fourth rows. $^*p < 0.05$, $^{**}p < 0.01$.)

| word-types | time-windows (ms) | | | | |
| | multi-level cleaning | | amplitude-based cleaning | | |
| | 650–750 | 700–800 | 200–300 | 650–750 | 700–800 |
|---|---|---|---|---|---|
| WORDS versus SIMILAR versus NONSENSE | * | * | * | ** | * |
| WORDS versus SIMILAR | n.s. | n.s. | n.s. | n.s. | n.s. |
| WORDS versus NONSENSE | * | n.s. | * | ** | * |
| | | | | interaction: * | |
| SIMILAR versus NONSENSE | * | * | * | n.s. | * |

## 3.5. Electroencephalography artefact rejection and analysis

EEG artefact rejection and analysis was done using the FIELDTRIP software package [43] in MATLAB R2017b. EEG data were digitally filtered off-line with a 0.01 Hz high-pass and 40 Hz low-pass filter and segmented between 200 ms pre-stimulus and 1000 ms after stimulus. Each segment was detrended and baselined between −200 ms (pre-stimulus) and 0 ms (stimulus onset).

For the multi-level cleaning procedure, amplitude-based artefact rejection was applied by removing each segment with amplitudes exceeding ±150 µV or with minimum and maximum values exceeding a 150 µV difference in 100 ms long sliding windows. Then, we coded the stimulus onsets and annotated the video-recordings using the ELAN software [44]. Segments were removed from the EEG data if 500 ms before and 1000 ms after the sound onset eye-, ear-, head- or any other body-movements (except of movements of breathing) were identified in the video-recordings. The remaining EEG segments were visually inspected for further artefacts (figure 1). For this, a horizontal EOG (hEOG) channel was created by a bipolar F7-F8 derivation. EOG channels (F7, F8 and hEOG) were inspected for blinks and horizontal eye-movements and for correlation with Fz and Cz channels. The person performing visual inspection of artefacts in the EEG data was blind to the experimental conditions of the segments. Based on the criteria used in EEG studies with infants (e.g. [23,36]), we accepted participants' data for further analysis if it contained a minimum of 10 artefact-free segments per condition and if the subject-average ERP had a flat pre-stimulus baseline with initially aligned ERPs across conditions. If these criteria were not fulfilled, data were rejected. We note that the criteria for minimum 10 trials might be too low for dogs who have a much thicker layer of muscles and skull and more fur on the top of their head. The mean number of trials per condition is shown in table 2 (left). Segments were averaged separately for each condition, participant and electrodes (Fz, Cz).

For the amplitude-based cleaning procedure, EEG data of the same participants whose data were used for analysis after the multi-level cleaning procedure were digitally filtered off-line with a 0.3 Hz high-pass and 40 Hz low-pass filter, and segmented between 200 ms pre-stimulus and 1000 ms after stimulus. Each segment was detrended and baselined between −200 ms and 0 ms. Amplitude-based artefact rejection was applied after detrending and baselining by removing each segment with amplitudes exceeding ±100 µV or with minimum and maximum values exceeding a 120 µV difference in 100 ms long sliding windows. The mean number of trials per condition is shown in table 2 (right). Segments were averaged separately for each condition, participant and electrodes (Fz, Cz).

To select time-windows of interest, we determined the onset and offset of possible effects by a 50 ms consecutive time-window analysis from 0 (word onset) to 1000 ms (similarly to [45]) with 100 ms long (see [28]) overlapping windows (e.g. between 0–100, 50–150 and 100–200 ms) using two-way (word-types and electrodes) repeated-measures ANOVAs in R [46–48] (table 3). We used a relatively long window (100 ms) because ERP effects were usually found in a few hundred milliseconds long time-window in human language studies (e.g. [17,28]). However, ERP responses can be also very quick; hence, differences between ERPs could be a result of more than one response. This inflates the possible effects of large data drifts. On the other hand, data averaged across larger time-windows are less sensitive for potential high-frequency noise.

Huynh–Feldt corrections were used for all the repeated-measures analyses with three-level factors. When word-types had a significant main effect or an interaction with electrodes in a time-window, we examined pairwise condition differences in a follow-up two-way repeated-measure ANOVAs as planned contrasts. When word-types had a significant effect in more than one consecutive time-windows in the comparison of the same pairs of word-types, we selected the beginning of the first significant time-window as onset and the end of the last significant time-window as offset. This procedure resulted in different time-windows for the pairs of word-types. We conducted the same sliding time-window analysis for the data cleaned with amplitude-based and with multi-level artefact rejection.

When time-windows were selected and there were differences between certain conditions in a time-window, we also tested the strength of evidence for no differences between the other conditions by the BF test described for ANOVA designs [24]. For this, we used the BAYESFACTOR package in R [49]. We tested the strength of evidence for the null hypothesis, i.e. the strength of evidence for no differences between conditions by comparing a model without and with word-type as a fixed effect. In both models, participant was a random effect. We report the BF with its proportional error.

Ethics. An ethical statement has been issued by the Hungarian 'Scientific and Ethical Committee of Animal Experiments' (PE/EA/853-2/2016). All owners volunteered to participate in the study and signed an informed consent form before the experiment.

Data accessibility. Stimulus material is available at https://doi.org/10.6084/m9.figshare.9807914.v2. Data are available at https://doi.org/10.6084/m9.figshare.9807758.v1.

Authors' contributions. Conceptualization: L.M. and A.A.; methodology: L.M.; investigation: L.M., T.A. and Zs.H.; data curation: T.A., Zs.H.; formal analysis: L.M., T.A., Zs.H.; writing—original draft: L.M. and Zs.H.; writing—reviewing and editing: L.M. and A.A.; funding acquisition: A.A.; supervision: A.A. and L.M.

Competing interests. The authors declare no competing interests.

Funding. The study was supported by the Hungarian Academy of Sciences via a grant to the MTA-ELTE 'Lendület' Neuroethology of Communication Research Group (grant no. LP2017-13/2017) and the Eötvös Loránd University. L.M. also received funding from National Research, Development and Innovation Office (NKFI, OTKA no. FK125417).

Acknowledgements. We are grateful to Nóra Bunford, Ádám Miklósi, Bálint Forgács, József Topál and Brigitta Tóth for their comments on earlier versions of the manuscript. We also thank Kálmán Czeibert for his help in providing CT images for the presentation of the electrode placement, Anna Kis, Márta Gácsi and Nóra Bunford for their advice on EEG methodology with dogs, and Adél Könczöl and Anna Gábor for their assistance in the experiments. We are also immensely grateful to all the owners that participated in this study with their dogs.

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
