## [Reviewer comments · Royal Society Open Science]

Review History

RSOS-191637.R0 (Original submission)

Review form: Reviewer 1 (Debbie Mills)

Is the manuscript scientifically sound in its present form?

Yes

Are the interpretations and conclusions justified by the results?

Yes

Is the language acceptable?

Yes

Do you have any ethical concerns with this paper?

No

Have you any concerns about statistical analyses in this paper?

No

Recommendation?

Accept with minor revision (please list in comments)

Comments to the Author(s)

Manuscript ID RSOS-191637, entitled "Event-related potentials reveal limited readiness to access phonetic details during word processing in dogs."

General comments:

The study addresses a theoretically interesting question about phonological perception of whole words in dogs. These findings are likely to be of interest to the general public as well as to developmental psychologists, cognitive neuroscientists, and researchers interested in animal cognition. Non-invasive event-related potentials (ERPs) were used to examine patterns of canine brain activity to perception of mispronunciations of familiar words. Since Pat Kuhl's work seminal work with chinchillas in the 1980s, scientists have known that a variety of non-human animals show robust categorical perception of human speech contrasts. Similar to the present study, non-invasive ERPs with young border collies revealed that dogs perceive human consonant/vowel speech sounds categorically (Adams, Molfese, & Betz, 1987 *Not cited in the current paper). However, this is the first study to examine brain activity to familiar words in dogs. The results suggest that dogs, like 14-month old human infants, treat mispronunciations of known words as acceptable instances of that word. The ERP results are strengthened by consistent findings in a behavioural experiment in dogs' perception of Hungarian words. The findings have implications for launching a plethora of studies examining cognitive development in dogs with implications for human language development.

Specific comments:

Overall the paper is well-written and a pleasure to read. Figures are very helpful and clearly illustrate the main points.

Stimuli. The stimuli for the mispronunciations were constructed by changing the vowel (ϵ vs ν for three stimuli; ϵ vs i for one) from a familiar word. In contrast, the nonsense words changed the initial consonant as well as other parts of the consonant/vowel strings. Hungarian has 14 vowels and if I am not mistaken, ϵ vs ν are close in physical distance between formants. This makes the mispronunciations much less perceptible than the nonsense words. A figure or table in supplemental material illustrating physical distance between vowel changes would be helpful. This is not a methodological problem, but it has implications for interpretation of the findings. Another difference between the mispronunciations and the nonsense words might be the frequency of the initial phonemes for the nonsense words. They are identical for the known words and mispronunciations. I'm not familiar with Hungarian, but if the initial sounds are less frequent it could account for increased perceptibility between known and nonsense words.

Another alternative interpretation of the results might be that dogs are sensitive to changes in the initial consonant (as changed in the nonsense words), but not vowels. This is unlikely for several reasons but should be discussed. First, Kuhl showed that in human infants, developmental changes in categorical perception of vowels precedes that of consonants. Second, although 14-month olds did not differentiate between known words and minimal pairs mispronunciations based on changing the initial consonant (Mills et al, 2004), a subsequent study (Mani, Mills, & Plunkett, 2102, DOI: 10.1111/j.1467-7687.2011.01092.x) showed that 14-month-olds did show ERP sensitivity to vowel mispronunciations when a pictorial context was provided. Third, in infants learning Hungarian, phonological perception of vowels occurs quite early in development e.g. see Gonzalez-Gomez, et al. 2019, <https://doi.org/10.1016/j.jecp.2018.08.014>. If dogs, like human infants, rely on distributional learning for phonological cues important for meaning, dogs exposed to Hungarian might be expected to pay attention to vowel changes. A brief but more in depth discussion of physical differences between stimuli is warranted.

EEG analysis.

Reference. Choice of a reference with dogs must be challenging due to a variety of factors. Using an active electrode site as the reference should have the same associated problems as it does in

human EEG research. Other EEG studies with dogs use a non-active common reference, e.g. on the nose. In the present study, ERP amplitudes at Pz would be subtracted from the other active sites, Therefore, it would make sense that ERP differences would be smaller at Cz than Fz because it is closer to Pz. Justification of the choice of Pz as the reference should be included in supplementary materials.

Artefact rejection. The procedure for artefact rejection was clearly presented and the method rigorous. According to Figure 1B & C, 24.87% were clean trials. That would be about 20 trials per condition. Yet, page 6 line 22 indicates there 59 trials per condition “One-way ANOVA showed no differences in the number of clean trials ($M=59$, $sd=10.57$) between conditions ($F(2,32)=1.143$, $p=.167$) (Fig.1C).” This is confusing and should be rewritten to reflect the number of trials per condition.

Additionally, the low number of trials per condition could be a concern. There were 80 trials per word type. In human infant ERP research, a recent study showed that when the number of trials per condition is low (i.e. 10-20 trials per condition), measurements and statistical outcomes of the resulting “clean” data can vary depending on the artefact treatment method chosen. The method of using the criteria of a minimum of 10 trials per condition might have been best practice a decade ago, but increasing the number of trials per condition to ensure replicability is of concern. One way to increase the number of trials might be to adjust the high pass filter settings (see next paragraph).

Filter settings. The filter settings of .01 to 40Hz are commendable. However, I wonder if a larger number of trials could be included if the off-line high pass filter was set to .1 or even .3 – neither of which should distort the data. More trials per condition would help with signal to noise ratio as well as help increase the potential for replicability.

Results.

The time windows were selected based on 100 ms moving windows (with 50 ms overlap) from 0 to 1000ms. This procedure is based on a previous study with dogs, and is a widely accepted method in human ERP studies. In the Mills et al. 2004, human infant study on which the current study is based, ERP amplitude differences were reported between 200 – 500 ms. Because the timing of ERP amplitude differences between dogs and humans is of particular interest, and because the ERP waveforms in the present study appear to show large amplitude differences at Fz from 200-500 ms, it would be helpful to include p values for those windows, even if it is in supplementary material. I’m sure that had those differences been significant, it would have been reported. But it would be nice to get an idea of where those apparent amplitude differences came from in terms of individual variability.

Discussion

Page 12 lines 15.16: “However, we also show here that important aspects of word processing are not shared between the two species. The ERP findings suggest that dogs’ word processing capacities are relatively slow and sensitive to the frequency with which dogs encounter the words.”

Actually, sensitivity to phonological and word frequency is something dogs have in common with humans across development. This has been shown in a large number of ERP studies as well as other behavioural studies. See papers below (just suggested not necessary to include these references – but frequency affects phonological and word processing in humans across the lifespan). Indeed, this is one of the main strengths of the paper! Figure 3 showing individual differences illustrating the ERP Word – nonsense ERP amplitudes is very impressive. Showing that these data replicate results from studies with human infants enhances the impact and believability of the study.

An important difference in phonological perception between dogs and human infants is likely to be on attention allocation. Early in development, human infants rely on domain general processes to pick up on statistical regularities (e.g. transitional probabilities within versus between words)

in the speech stream to segment words from continuous speech. With increased experience infants learn to use more domain-specific information such as stress and prosodic cues such as infant-directed speech. On the other hand, dogs are more likely to learn isolated words as commands associated with food or social rewards. What is so remarkable about the present study are the similarities, not the differences, between dogs and human infants in the way the brain processes familiar words.

Phonological development:

Sita Minke ter Haar & Clara Cecilia Levelt (2018) Disentangling Attention for Frequency and Phonological Markedness in 9- and 12-Month-Old Infants, *Language Learning and Development*, 14:4, 279-296, DOI: 10.1080/15475441.2018.1480375

Swingley D. (2009). Contributions of infant word learning to language development. *Philosophical transactions of the Royal Society of London. Series B, Biological sciences*, 364(1536), 3617-3632. doi:10.1098/rstb.2009.0107

Experience with individual words:

Mills, D. L., Plunkett, K., Prat, C., & Schafer, G. (2005). Watching the infant brain learn words: Effects of language and experience. *Cognitive Development*, 20, 19-31.

Minor comments:

Add reference below to sentence: "Fully non-invasive EEG has been applied in earlier studies on awake but trained dogs (17,18)."

Christina L. Adams, Dennis L. Molfese & Jacqueline C. Betz (1987) Electrophysiological correlates of categorical speech perception for voicing contrasts in dogs, *Developmental Neuropsychology*, 3:3-4, 175-189, DOI: 10.1080/87565648709540375

Review form: Reviewer 2

Is the manuscript scientifically sound in its present form?

No

Are the interpretations and conclusions justified by the results?

No

Is the language acceptable?

Yes

Do you have any ethical concerns with this paper?

No

Have you any concerns about statistical analyses in this paper?

Yes

Recommendation?

Reject

Comments to the Author(s)

Please see the full review as an attachment (Appendix A).

Decision letter (RSOS-191637.R0)

12-Dec-2019

Dear Dr Magyari:

Manuscript ID RSOS-191637 entitled "Event-related potentials reveal limited readiness to access phonetic details during word processing in dogs" which you submitted to Royal Society Open Science, has been reviewed. The comments from reviewers are included at the bottom of this letter.

In view of the criticisms of the reviewers, the manuscript has been rejected in its current form. However, a new manuscript may be submitted which takes into consideration these comments.

Please note that resubmitting your manuscript does not guarantee eventual acceptance, and that your resubmission will be subject to peer review before a decision is made.

Your resubmitted manuscript should be submitted by 10-Jun-2020. If you are unable to submit by this date please contact the Editorial Office.

Kind regards,
Anita Kristiansen
Editorial Coordinator
Royal Society Open Science
openscience@royalsociety.org

on behalf of Dr César Lima (Associate Editor) and Essi Viding (Subject Editor)
openscience@royalsociety.org

Reviewers' Comments to Author:
Reviewer: 1

Comments to the Author(s)
Manuscript ID RSOS-191637, entitled "Event-related potentials reveal limited readiness to access phonetic details during word processing in dogs."

General comments:

The study addresses a theoretically interesting question about phonological perception of whole words in dogs. These findings are likely to be of interest to the general public as well as to developmental psychologists, cognitive neuroscientists, and researchers interested in animal cognition. Non-invasive event-related potentials (ERPs) were used to examine patterns of canine brain activity to perception of mispronunciations of familiar words. Since Pat Kuhl's work seminal work with chinchillas in the 1980s, scientists have known that a variety of non-human

animals show robust categorical perception of human speech contrasts. Similar to the present study, non-invasive ERPs with young border collies revealed that dogs perceive human consonant/vowel speech sounds categorically (Adams, Molfese, & Betz, 1987 *Not cited in the current paper). However, this is the first study to examine brain activity to familiar words in dogs. The results suggest that dogs, like 14-month old human infants, treat mispronunciations of known words as acceptable instances of that word. The ERP results are strengthened by consistent findings in a behavioural experiment in dogs' perception of Hungarian words. The findings have implications for launching a plethora of studies examining cognitive development in dogs with implications for human language development.

Specific comments:

Overall the paper is well-written and a pleasure to read. Figures are very helpful and clearly illustrate the main points.

Stimuli. The stimuli for the mispronunciations were constructed by changing the vowel (ϵ vs ν for three stimuli; ϵ vs i for one) from a familiar word. In contrast, the nonsense words changed the initial consonant as well as other parts of the consonant/vowel strings. Hungarian has 14 vowels and if I am not mistaken, ϵ vs ν are close in physical distance between formants. This makes the mispronunciations much less perceptible than the nonsense words. A figure or table in supplemental material illustrating physical distance between vowel changes would be helpful. This is not a methodological problem, but it has implications for interpretation of the findings. Another difference between the mispronunciations and the nonsense words might be the frequency of the initial phonemes for the nonsense words. They are identical for the known words and mispronunciations. I'm not familiar with Hungarian, but if the initial sounds are less frequent it could account for increased perceptibility between known and nonsense words.

Another alternative interpretation of the results might be that dogs are sensitive to changes in the initial consonant (as changed in the nonsense words), but not vowels. This is unlikely for several reasons but should be discussed. First, Kuhl showed that in human infants, developmental changes in categorical perception of vowels precedes that of consonants. Second, although 14-month olds did not differentiate between known words and minimal pairs mispronunciations based on changing the initial consonant (Mills et al, 2004), a subsequent study (Mani, Mills, & Plunkett, 2102, DOI: 10.1111/j.1467-7687.2011.01092.x) showed that 14-month-olds did show ERP sensitivity to vowel mispronunciations when a pictorial context was provided. Third, in infants learning Hungarian, phonological perception of vowels occurs quite early in development e.g. see Gonzalez-Gomez, et al. 2019, <https://doi.org/10.1016/j.jecp.2018.08.014>. If dogs, like human infants, rely on distributional learning for phonological cues important for meaning, dogs exposed to Hungarian might be expected to pay attention to vowel changes. A brief but more in depth discussion of physical differences between stimuli is warranted.

EEG analysis.

Reference. Choice of a reference with dogs must be challenging due to a variety of factors. Using an active electrode site as the reference should have the same associated problems as it does in human EEG research. Other EEG studies with dogs use a non-active common reference, e.g. on the nose. In the present study, ERP amplitudes at Pz would be subtracted from the other active sites, Therefore, it would make sense that ERP differences would be smaller at Cz than Fz because it is closer to Pz. Justification of the choice of Pz as the reference should be included in supplementary materials.

Artefact rejection. The procedure for artefact rejection was clearly presented and the method rigorous. According to Figure 1B & C, 24.87% were clean trials. That would be about 20 trials per condition. Yet, page 6 line 22 indicates there 59 trials per condition "One-way ANOVA showed no differences in the number of clean trials ($M=59$, $sd=10.57$) between conditions ($F(2,32)=1.143$, $p=.167$) (Fig.1C)." This is confusing and should be rewritten to reflect the number of trials per condition.

Additionally, the low number of trials per condition could be a concern. There were 80 trials per word type. In human infant ERP research, a recent study showed that when the number of trials per condition is low (i.e. 10-20 trials per condition), measurements and statistical outcomes of the resulting “clean” data can vary depending on the artefact treatment method chosen. The method of using the criteria of a minimum of 10 trials per condition might have been best practice a decade ago, but increasing the number of trials per condition to ensure replicability is of concern. One way to increase the number of trials might be to adjust the high pass filter settings (see next paragraph).

Filter settings. The filter settings of .01 to 40Hz are commendable. However, I wonder if a larger number of trials could be included if the off-line high pass filter was set to .1 or even .3 – neither of which should distort the data. More trials per condition would help with signal to noise ratio as well as help increase the potential for replicability.

Results.

The time windows were selected based on 100 ms moving windows (with 50 ms overlap) from 0 to 1000ms. This procedure is based on a previous study with dogs, and is a widely accepted method in human ERP studies. In the Mills et al. 2004, human infant study on which the current study is based, ERP amplitude differences were reported between 200 – 500 ms. Because the timing of ERP amplitude differences between dogs and humans is of particular interest, and because the ERP waveforms in the present study appear to show large amplitude differences at Fz from 200-500 ms, it would be helpful to include p values for those windows, even if it is in supplementary material. I’m sure that had those differences been significant, it would have been reported. But it would be nice to get an idea of where those apparent amplitude differences came from in terms of individual variability.

Discussion

Page 12 lines 15.16: “However, we also show here that important aspects of word processing are not shared between the two species. The ERP findings suggest that dogs’ word processing capacities are relatively slow and sensitive to the frequency with which dogs encounter the words.”

Actually, sensitivity to phonological and word frequency is something dogs have in common with humans across development. This has been shown in a large number of ERP studies as well as other behavioural studies. See papers below (just suggested not necessary to include these references – but frequency affects phonological and word processing in humans across the lifespan). Indeed, this is one of the main strengths of the paper! Figure 3 showing individual differences illustrating the ERP Word – nonsense ERP amplitudes is very impressive. Showing that these data replicate results from studies with human infants enhances the impact and believability of the study.

An important difference in phonological perception between dogs and human infants is likely to be on attention allocation. Early in development, human infants rely on domain general processes to pick up on statistical regularities (e.g. transitional probabilities within versus between words) in the speech stream to segment words from continuous speech. With increased experience infants learn to use more domain-specific information such as stress and prosodic cues such as infant-directed speech. On the other hand, dogs are more likely to learn isolated words as commands associated with food or social rewards. What is so remarkable about the present study are the similarities, not the differences, between dogs and human infants in the way the brain processes familiar words.

Phonological development:

Sita Minke ter Haar & Clara Cecilia Levelt (2018) Disentangling Attention for Frequency and Phonological Markedness in 9- and 12-Month-Old Infants, *Language Learning and Development*, 14:4, 279-296, DOI: 10.1080/15475441.2018.1480375

Swingley D. (2009). Contributions of infant word learning to language development. *Philosophical transactions of the Royal Society of London. Series B, Biological sciences*, 364(1536), 3617–3632. doi:10.1098/rstb.2009.0107

Experience with individual words:

Mills, D. L., Plunkett, K., Prat, C., & Schafer, G. (2005). Watching the infant brain learn words: Effects of language and experience. *Cognitive Development*, 20, 19-31.

Minor comments:

Add reference below to sentence: “Fully non-invasive EEG has been applied in earlier studies on awake but trained dogs (17,18).”

Christina L. Adams, Dennis L. Molfese & Jacqueline C. Betz (1987) Electrophysiological correlates of categorical speech perception for voicing contrasts in dogs, *Developmental Neuropsychology*, 3:3-4, 175-189, DOI: 10.1080/87565648709540375

Reviewer: 2

Comments to the Author(s)

Please see the full review as an attachment. (RSOS_review_11_2019.pdf)

Author's Response to Decision Letter for (RSOS-191637.R0)

See Appendix B.

RSOS-200851.R0

Review form: Reviewer 1 (Debbie Mills)

Is the manuscript scientifically sound in its present form?

Yes

Are the interpretations and conclusions justified by the results?

Yes

Is the language acceptable?

Yes

Do you have any ethical concerns with this paper?

No

Have you any concerns about statistical analyses in this paper?

No

Recommendation?

Accept with minor revision (please list in comments)

Comments to the Author(s)

The authors are to be commended for their careful response to methodological concerns regarding the stimuli and data analysis. As this study is one of the first of its kind and might be used as the methodological standard for subsequent research. Thus, it is important to stress methodological rigour. The frequency table for the initial phonemes and addressing the possibility of physical differences between the mispronunciations and nonsense words in the discussion were helpful. Examination of the effects of different approaches to artefact treatment on statistical analyses across different time windows, has important methodological implications for EEG research with human infants as well. As a reader who is interested in EEG methodology, I found this quite engaging. However, I wonder if the typical RSOS reader would find the article easier to process if only the one -step analysis was presented – and the comparisons of different artefact treatments were presented in the supplemental materials.

My only other comments are in regard to the conclusion that dogs show limited phonological processing. For example a) “the temporal dynamics of word processing in dogs might be comparable to that in humans but dog’s capacity to access phonological detail is more limited” (abstract), Also, b) “These findings suggest that reduced readiness to process phonetic details may be one reason that incapacitates dogs from acquiring a sizeable vocabulary” (conclusion). I agree those statements are both likely to be true, but the authors didn’t demonstrate how dogs differed from human infants based on their data. Unless they can explicitly state how the ERP data show limited access to phonological detail, those ideas are speculative and for future research.

Of particular interest was the correlation between the owner’s word use frequency and the size of the ERP amplitude difference in the late but not early time window. It is possible, that like human infants these time windows reflect qualitatively different aspects of word processing in dogs. Although, as the statistical significance ERP differences in this time window differed depending on the artefact rejection method, it is commendable not to be too speculative. Overall, this approach leads the way to examine many different theoretical questions on non-human sensitivity to the phonology of human speech.

Review form: Reviewer 2

Is the manuscript scientifically sound in its present form?

No

Are the interpretations and conclusions justified by the results?

No

Is the language acceptable?

Yes

Do you have any ethical concerns with this paper?

No

Have you any concerns about statistical analyses in this paper?

No

Recommendation?

Major revision is needed (please make suggestions in comments)

Comments to the Author(s)

Please see the attached review file (Appendix C).

Decision letter (RSOS-200851.R0)

Dear Dr Magyari,

The Subject Editor assigned to your paper ("Event-related potentials reveal limited readiness to access phonetic details during word processing in dogs") has now received comments from reviewers. We would like you to revise your paper in accordance with the referee and Associate Editor suggestions which can be found below (not including confidential reports to the Editor). Please note this decision does not guarantee eventual acceptance.

Please submit a copy of your revised paper before 30-Jul-2020. Please note that the revision deadline will expire at 00.00am on this date. If we do not hear from you within this time then it will be assumed that the paper has been withdrawn. In exceptional circumstances, extensions may be possible if agreed with the Editorial Office in advance. We do not allow multiple rounds of revision so we urge you to make every effort to fully address all of the comments at this stage. If deemed necessary by the Editors, your manuscript will be sent back to one or more of the original reviewers for assessment. If the original reviewers are not available we may invite new reviewers.

When submitting your revised manuscript, you must respond to the comments made by the referees and upload a file "Response to Referees" in "Section 6 - File Upload". Please use this to document how you have responded to each of the comments, and the adjustments you have made. In order to expedite the processing of the revised manuscript, please be as specific as possible in your response.

- Ethics statement

- Data accessibility

If you wish to submit your supporting data or code to Dryad (<http://datadryad.org/>), or modify your current submission to dryad, please use the following link:
<http://datadryad.org/submit?journalID=RSOS&manu=RSOS-200851>

- Competing interests

- Authors' contributions

- Acknowledgements

- Funding statement

Kind regards,
 Lianne Parkhouse
 Editorial Coordinator
 Royal Society Open Science
openscience@royalsociety.org

on behalf of Dr César Lima (Associate Editor) and Essi Viding (Subject Editor)
openscience@royalsociety.org

Associate Editor Comments to Author (Dr César Lima):

We invite the authors to address the remaining concerns in a major revision, paying particular attention to the power issues raised by Reviewer 2. This shortcoming should be explicitly acknowledged and discussed, and the N should be included in the abstract.

As noted by Reviewer 1, inferences based on indirect comparisons (between dogs and humans) also need to be made more cautiously, and clearly phrased as speculative and for future work. The same applies to inferences based on null results, as these do not provide evidence for the null hypothesis (e.g., processing similarities between known words and similar nonsense words). In

fact, running Bayesian analyses would be ideal if the authors want to formally interpret these findings.

Finally, the authors should also expand on the methodological contribution of their work.

Reviewer comments to Author:

Reviewer: 1

Comments to the Author(s)

The authors are to be commended for their careful response to methodological concerns regarding the stimuli and data analysis. As this study is one of the first of its kind and might be used as the methodological standard for subsequent research. Thus, it is important to stress methodological rigour. The frequency table for the initial phonemes and addressing the possibility of physical differences between the mispronunciations and nonsense words in the discussion were helpful. Examination of the effects of different approaches to artefact treatment on statistical analyses across different time windows, has important methodological implications for EEG research with human infants as well. As a reader who is interested in EEG methodology, I found this quite engaging. However, I wonder if the typical RSOS reader would find the article easier to process if only the one -step analysis was presented – and the comparisons of different artefact treatments were presented in the supplemental materials.

My only other comments are in regard to the conclusion that dogs show limited phonological processing. For example a) “the temporal dynamics of word processing in dogs might be comparable to that in humans but dog’s capacity to access phonological detail is more limited” (abstract), Also, b) “These findings suggest that reduced readiness to process phonetic details may be one reason that incapacitates dogs from acquiring a sizeable vocabulary” (conclusion). I agree those statements are both likely to be true, but the authors didn’t demonstrate how dogs differed from human infants based on their data. Unless they can explicitly state how the ERP data show limited access to phonological detail, those ideas are speculative and for future research.

Of particular interest was the correlation between the owner’s word use frequency and the size of the ERP amplitude difference in the late but not early time window. It is possible, that like human infants these time windows reflect qualitatively different aspects of word processing in dogs. Although, as the statistical significance ERP differences in this time window differed depending on the artefact rejection method, it is commendable not to be too speculative. Overall, this approach leads the way to examine many different theoretical questions on non-human sensitivity to the phonology of human speech.

Reviewer: 2

Comments to the Author(s)

Please see the attached review file.

Author's Response to Decision Letter for (RSOS-200851.R0)

See Appendix D.

RSOS-200851.R1 (Revision)

Review form: Reviewer 1 (Debbie Mills)

Is the manuscript scientifically sound in its present form?

Yes

Are the interpretations and conclusions justified by the results?

Yes

Is the language acceptable?

Yes

Do you have any ethical concerns with this paper?

No

Have you any concerns about statistical analyses in this paper?

No

Recommendation?

Accept as is

Comments to the Author(s)

The Editor's letter suggested using bayesian analyses to support the position that ERPs to known and phonologically similar words did not differ. I find this a useful approach, and agree it would be helpful. This is especially relevant as the paper is being showcased for its methodological rigour and approach. Otherwise, I am happy with the revisions.

Review form: Reviewer 2

Is the manuscript scientifically sound in its present form?

No

Are the interpretations and conclusions justified by the results?

No

Is the language acceptable?

Yes

Do you have any ethical concerns with this paper?

No

Have you any concerns about statistical analyses in this paper?

Yes

Recommendation?

Major revision is needed (please make suggestions in comments)

Comments to the Author(s)

Magyari et al, RSOS-200851.R1 "Event-related potentials reveal limited readiness to access phonetic details during word processing in dogs"

The revised work by Magyari et al has now improved with the more cautious interpretation of the data, including the grand-average across conditions situations adding to the replicability of the study and by considering the possible artifacts more carefully. I am glad that the authors have now considered their contribution on a grander scale for the field, how their input piece fits into the big picture. I think this will be an informative contribution to this growing field. I have now gone through the manuscript now in more detail, and I have the following further suggestions or concerns:

Introduction, lines 65-72: "Here we study... - ...method (needles inserted under the skin) (20)."

-> Continuous EEG of dogs have been abundantly measured in the context of disorders such as epilepsy, which could be mentioned. Still, there has been a long way from continuous EEG to event-related potentials. I think this paragraph now includes the most relevant background work on the development of dog ERP measurement, but it appears rather dismissive on this previous groundwork. As the method is still quite young in dogs, the accumulative information is needed for the field. This section would deserve to be explained in a bit more detail, opening the contribution of these papers for the field and for the present paper. You have succeeded in conducting first scalp-EEG auditory measurements of non-medicated dogs: this is a great achievement, and likely, did not come easily.

Introduction (ln 45-46): "Studies have recently also shown similarities in the neural correlates of human and dog word processing (12,13)."

-> Ref 13 does not involve neural correlates nor neuroscientific methods, thus should be introduced as a behavioral measure. The authors could also indicate that ref 12 is coming from the same lab. This is both a praise for the group and a sign for the need to replicate the findings on a global level.

Intro (58-59) "Younger infants (around 14 months) fail to associate phonetically similar words such as bih or dih to different objects in word learning situations (15,16)."

-> As many of the readers do not know the meaning of bih or dih, the authors could indicate the language and English translation of these.

lines 89-96 and throughout the text: The nomenclature used for the analysis (automatic artifact-rejection: one-step cleaning and a multi-level method for artifact-rejection: three-step cleaning) are non-standard EEG terminology and somewhat misleading in the current form. I suggest calling the simple epoch rejection based on amplitude value not as "automatic", but as amplitude-based artifact rejection. Also, what is rigorous is not straightforward but open to personal opinions, so I suggest changing the wording of the following sentence accordingly, based on facts:

"According to our knowledge, such a rigorous method has been applied here for the first time on the EEG of awake dogs."

-> "According to our knowledge, combining visual data exclusion with amplitude-based artifact rejection has been applied here for the first time on the EEG of awake dogs."

lines 120-125: The grand average across conditions in fig s2 is extremely helpful and raises both credibility and reproducibility of the current study, I suggest including it as one of the figures of the manuscript instead of being in supplementary material, where it easily can be lost from most readers. However, it is not advisable to refer to frequency filter as data cleaning procedure here, as it is merely a filter. So, "For better visualization of the general shape of the ERP, we also conducted a third data-cleaning procedure. This was similar to the one-step data cleaning procedure explained before with the exception of the low-pass filter which was set to a lower value (20 Hz)."

-> change e.g. "For better visualization of the general shape of the ERP and easier detection of e.g. the primary auditory responses of the data, we set the low-pass filter to a lower value (20Hz)."

From the fig s2 we can detect the 100 ms primary auditory -like response, which is a big relief. The fact that the 200-ms response appears even larger is puzzling, since the primary responses are usually the most robust of EEG responses. Sometimes the responses appear a bit oscillatory, though, so this might be that. It would be interesting to see these compared to a pure auditory response, with stimulus such as sine wave. This would confirm whether the 200-300 ms responses have anything to do with speech sounds at all or whether this is some kind of oscillatory echo of the earlier response: these are dogs and we are still at the early days. I suggest adding a little bit more explanation on this part, so that the reader knows the value of this step (confirms that we actually do appear to get a primary auditory-like response of dogs with this method and adds to the reproducibility of the step). This is very important for those who know EEG methodology, for the credibility of the whole field.

In your response letter, you had a reasonable discussion of this issue, I suggest adding it (perhaps slightly modified) also in your manuscript: "In our study, future research could address whether the observed small negativity around 100 ms is similar to an N1 ERP component exhibited by humans. Our experiment was not designed for revealing the N1 component (i.e. we did not expect any modulation of this component), hence, we cannot make any conclusion about it. Future studies could also examine whether the negativity around 200-300 ms is stimulus-specific or also reflects a more general auditory response."

lines 260-262: "Moreover, even if ERP effects were due to muscle movements, those would still show that dogs differentiate word-categories, hence, such a confound would not undermine the claim about dogs' ability for discriminating known words from nonsense words."

-> I agree with this -we don't need a laborious EEG study to know that dogs differentiate human words, we know they do on the basis of behavior only. However this discussion is important in determining the neural processing: whether we have a window into it with the data or not, HOW is it that dogs process the human words. This is a difference worth mentioning.

Methods, line 457 onward: The authors used a 100ms long sliding window for the statistical analysis, in steps of 50 ms. This is a very long time window in the EEG analysis, since ERP responses can be very quick. In practice, this means that within that time window, the differences calculated can be sums from more than one response (e.g. one positive and subsequent negative potential). This inflates the possible effects of large data drifts, but on the other hand can help to overcome potential hf noise. It is good to discuss the effects of the analysis choices - there is always a trade-off.

Results&Discussion, ln 118-119 and Methods, ln 478-480 "When significant differences between conditions were found in a time-window, we also tested condition-differences by a non-parametric statistical test (Wilcoxon-signed rank test)." and "Condition differences at selected time-windows were also tested by Wilcoxon signed-rank test. When the repeated-measures ANOVA did not show any interaction effect between channels and condition in a time-window, Wilcoxon signed-rank test was applied on the mean of Fz and Cz. When there was an interaction effect and the post-hoc tests showed an effect only at one of the electrodes, Wilcoxon signed-rank test was calculated for the values of this one electrode."

-> I do not understand the use of Wilcoxon test as a post-hoc for repeated-measured ANOVA here. It is overly liberal for the normal data, Student's t test in its stead is advisable for the normally distributed data.

Decision letter (RSOS-200851.R1)

Dear Dr Magyari

The Editors assigned to your paper RSOS-200851.R1 "Event-related potentials reveal limited readiness to access phonetic details during word processing in dogs" have now received comments from reviewers and would like you to revise the paper in accordance with the reviewer comments and any comments from the Editors. Please note this decision does not guarantee eventual acceptance.

We do not generally allow multiple rounds of revision, so it is unusual the Editors have offered you a further round here. Please note that if you are not able to satisfy the referees and Editors that the paper is ready for acceptance, the paper will likely be rejected. If deemed necessary by the Editors, your manuscript will be sent back to one or more of the original reviewers for assessment. If the original reviewers are not available, we may invite new reviewers.

Please submit your revised manuscript and required files (see below) no later than 21 days from today's (ie 15-Sep-2020) date. Note: the ScholarOne system will 'lock' if submission of the revision is attempted 21 or more days after the deadline. If you do not think you will be able to meet this deadline please contact the editorial office immediately.

on behalf of Dr César Lima (Associate Editor) and Essi Viding (Subject Editor)
openscience@royalsociety.org

Reviewer comments to Author:
Reviewer: 2

Comments to the Author(s)
Magyari et al, RSOS-200851.R1 "Event-related potentials reveal limited readiness to access phonetic details during word processing in dogs"

The revised work by Magyari et al has now improved with the more cautious interpretation of the data, including the grand-average across conditions situations adding to the replicability of the study and by considering the possible artifacts more carefully. I am glad that the authors have now considered their contribution on a grander scale for the field, how their input piece fits into

the big picture. I think this will be an informative contribution to this growing field. I have now gone through the manuscript now in more detail, and I have the following further suggestions or concerns:

Introduction, lines 65-72: "Here we study... - ...method (needles inserted under the skin) (20)."
 -> Continuous EEG of dogs have been abundantly measured in the context of disorders such as epilepsy, which could be mentioned. Still, there has been a long way from continuous EEG to event-related potentials. I think this paragraph now includes the most relevant background work on the development of dog ERP measurement, but it appears rather dismissive on this previous groundwork. As the method is still quite young in dogs, the accumulative information is needed for the field. This section would deserve to be explained in a bit more detail, opening the contribution of these papers for the field and for the present paper. You have succeeded in conducting first scalp-EEG auditory measurements of non-medicated dogs: this is a great achievement, and likely, did not come easily.

Introduction (ln 45-46): "Studies have recently also shown similarities in the neural correlates of human and dog word processing (12,13)."

-> Ref 13 does not involve neural correlates nor neuroscientific methods, thus should be introduced as a behavioral measure. The authors could also indicate that ref 12 is coming from the same lab. This is both a praise for the group and a sign for the need to replicate the findings on a global level.

Intro (58-59) "Younger infants (around 14 months) fail to associate phonetically similar words such as bih or dih to different objects in word learning situations (15,16)."

-> As many of the readers do not know the meaning of bih or dih, the authors could indicate the language and English translation of these.

lines 89-96 and throughout the text: The nomenclature used for the analysis (automatic artifact-rejection: one-step cleaning and a multi-level method for artifact-rejection: three-step cleaning) are non-standard EEG terminology and somewhat misleading in the current form. I suggest calling the simple epoch rejection based on amplitude value not as "automatic", but as amplitude-based artifact rejection. Also, what is rigorous is not straightforward but open to personal opinions, so I suggest changing the wording of the following sentence accordingly, based on facts:

"According to our knowledge, such a rigorous method has been applied here for the first time on the EEG of awake dogs."

-> "According to our knowledge, combining visual data exclusion with amplitude-based artifact rejection has been applied here for the first time on the EEG of awake dogs."

lines 120-125: The grand average across conditions in fig s2 is extremely helpful and raises both credibility and reproducibility of the current study, I suggest including it as one of the figures of the manuscript instead of being in supplementary material, where it easily can be lost from most readers. However, it is not advisable to refer to frequency filter as data cleaning procedure here, as it is merely a filter. So, "For better visualization of the general shape of the ERP, we also conducted a third data-cleaning procedure. This was similar to the one-step data cleaning procedure explained before with the exception of the low-pass filter which was set to a lower value (20 Hz)."

-> change e.g. "For better visualization of the general shape of the ERP and easier detection of e.g. the primary auditory responses of the data, we set the low-pass filter to a lower value (20Hz)."

From the fig s2 we can detect the 100 ms primary auditory -like response, which is a big relief. The fact that the 200-ms response appears even larger is puzzling, since the primary responses are usually the most robust of EEG responses. Sometimes the responses appear a bit oscillatory, though, so this might be that. It would be interesting to see these compared to a pure auditory response, with stimulus such as sine wave. This would confirm whether the 200-300 ms

responses have anything to do with speech sounds at all or whether this is some kind of oscillatory echo of the earlier response: these are dogs and we are still at the early days. I suggest adding a little bit more explanation on this part, so that the reader knows the value of this step (confirms that we actually do appear to get a primary auditory-like response of dogs with this method and adds to the reproducibility of the step). This is very important for those who know EEG methodology, for the credibility of the whole field.

In your response letter, you had a reasonable discussion of this issue, I suggest adding it (perhaps slightly modified) also in your manuscript: "In our study, future research could address whether the observed small negativity around 100 ms is similar to an N1 ERP component exhibited by humans. Our experiment was not designed for revealing the N1 component (i.e. we did not expect any modulation of this component), hence, we cannot make any conclusion about it. Future studies could also examine whether the negativity around 200-300 ms is stimulus-specific or also reflects a more general auditory response."

lines 260-262: "Moreover, even if ERP effects were due to muscle movements, those would still show that dogs differentiate word-categories, hence, such a confound would not undermine the claim about dogs' ability for discriminating known words from nonsense words."

-> I agree with this -we don't need a laborious EEG study to know that dogs differentiate human words, we know they do on the basis of behavior only. However this discussion is important in determining the neural processing: whether we have a window into it with the data or not, HOW is it that dogs process the human words. This is a difference worth mentioning.

Methods, line 457 onward: The authors used a 100ms long sliding window for the statistical analysis, in steps of 50 ms. This is a very long time window in the EEG analysis, since ERP responses can be very quick. In practice, this means that within that time window, the differences calculated can be sums from more than one response (e.g. one positive and subsequent negative potential). This inflates the possible effects of large data drifts, but on the other hand can help to overcome potential hf noise. It is good to discuss the effects of the analysis choices - there is always a trade-off.

Results&Discussion, ln 118-119 and Methods, ln 478-480 "When significant differences between conditions were found in a time-window, we also tested condition-differences by a non-parametric statistical test (Wilcoxon-signed rank test)." and "Condition differences at selected time-windows were also tested by Wilcoxon signed-rank test. When the repeated-measures ANOVA did not show any interaction effect between channels and condition in a time-window, Wilcoxon signed-rank test was applied on the mean of Fz and Cz. When there was an interaction effect and the post-hoc tests showed an effect only at one of the electrodes, Wilcoxon signed-rank test was calculated for the values of this one electrode."

-> I do not understand the use of Wilcoxon test as a post-hoc for repeated-measured ANOVA here. It is overly liberal for the normal data, Student's t test in its stead is advisable for the normally distributed data.

Reviewer: 1

Comments to the Author(s)

The Editor's letter suggested using bayesian analyses to support the position that ERPs to known and phonologically similar words did not differ. I find this a useful approach, and agree it would be helpful. This is especially relevant as the paper is being showcased for its methodological rigour and approach. Otherwise, I am happy with the revisions.

===PREPARING YOUR MANUSCRIPT===

===PREPARING YOUR REVISION IN SCHOLARONE===

- If you are providing image files for potential cover images, please upload these at this step, and inform the editorial office you have done so. You must hold the copyright to any image provided.
- A copy of your point-by-point response to referees and Editors. This will expedite the preparation of your proof.

- Ensure that your data access statement meets the requirements at <https://royalsociety.org/journals/authors/author-guidelines/#data>. You should ensure that you cite the dataset in your reference list. If you have deposited data etc in the Dryad repository, please include both the 'For publication' link and 'For review' link at this stage.
- If you are requesting an article processing charge waiver, you must select the relevant waiver option (if requesting a discretionary waiver, the form should have been uploaded at Step 3 'File upload' above).
- If you have uploaded ESM files, please ensure you follow the guidance at <https://royalsociety.org/journals/authors/author-guidelines/#supplementary-material> to include a suitable title and informative caption. An example of appropriate titling and captioning may be found at https://figshare.com/articles/Table_S2_from_Is_there_a_trade-off_between_peak_performance_and_performance_breadth_across_temperatures_for_aerobic_scope_in_teleost_fishes_/3843624.

Author's Response to Decision Letter for (RSOS-200851.R1)

See Appendix E.

RSOS-200851.R2 (Revision)

Review form: Reviewer 2

Is the manuscript scientifically sound in its present form?

Yes

Are the interpretations and conclusions justified by the results?

Yes

Is the language acceptable?

Yes

Do you have any ethical concerns with this paper?

No

Have you any concerns about statistical analyses in this paper?

No

Recommendation?

Accept as is

Comments to the Author(s)

My previous major concerns have been adequately addressed.

Decision letter (RSOS-200851.R2)

Dear Dr Magyari,

It is a pleasure to accept your manuscript entitled "Event-related potentials reveal limited readiness to access phonetic details during word processing in dogs" in its current form for publication in Royal Society Open Science. The comments of the reviewer(s) who reviewed your manuscript are included at the foot of this letter.

on behalf of Dr César Lima (Associate Editor) and Essi Viding (Subject Editor)
openscience@royalsociety.org

Reviewer comments to Author:
Reviewer: 2

Comments to the Author(s)

My previous major concerns have been adequately addressed.

Appendix A

“Event-related potentials reveal limited readiness to access phonetic details during word processing in dogs”

by Magyari, Huszár, Turzó, and Andics

The article by Magyari and colleagues is of a highly timely topic and the authors, together with possible research assistants, have obviously put considerable amounts of work and effort into this manuscript. Personally, I would love to see more research on this topic. However, the first studies of this kind need to be absolutely trustworthy, since all the forthcoming related science will rely on them. I am sad to say that this manuscript does not meet the quality standards that we need for this kind of work studying neuroscientific basis of dog cognition.

My main concern in this paper is the sufficiency and the quality of the data that all the discussion is based on. I am seriously concerned that the results reported in the paper are due to false positive findings, and they are not reproducible, for the following reasons.

The authors report that they have obtained approximately 20 clean event-related responses of each of the three auditory conditions per dog. They state “mean number of clean trials (y-axis) per condition (x-axis)” on the Fig. 1 legend (page 8/lines 93-94), and the Figure 1 y-axis values approximate +/- 20. Taken the signal-to-noise ratio of electroencephalography into account, obtaining a difference of event-related potentials to 20+20+20 stimuli is highly unreliable also in human EEG data, and the SNR of the dog EEG data is far worse due to the signal decay over distance and the conductance differences in the tissues between the signal source and the measurement site. Also, the impedances of dog EEG tend to be worse than those of human EEG, lowering the SNR. Also in this study, the impedances are reported to be 15 or lower. Thus, the event-related samples of approx. 20 trials per dog per condition, with n=17 participant dogs' data included, are inadequate to reach any final conclusion.

Furthermore, there is no mention of background noise conditions in the room where the recordings take place. Was the recording room properly soundproof, as this is an auditory study? Were there any ambient background noise in the room? Was anyone else present than the researcher, owner and the dog? Did the dog stay still during the measurements?

The placing of electrodes would also need more justification, as there are no common standards yet in the dog neuroscientific research. Why did the authors decide to measure the data from these locations? How did the authors decide the location of the reference electrode? Why this number of electrodes? And finally, how did the authors take the previous dog ERP research into consideration when deciding these?

Images of event-related responses on individual dogs would be also very informative, and the numbers of trials that reached analysis from each of the participant dogs. From the current data, it is impossible to determine e.g. if the Grand Average response depicted in Fig. 2A is mainly due to a few dogs' leading effect, or whether the GA indeed represents a valuable average of the whole population. In addition, standard error levels should be added to the Fig 2A, just to clarify how much of the response can be distinguished from the noise levels. Looking at the Fig. 2A as it is now, one could deduce that there is a possibility of an effect of the words vs. nonsense conditions around 200ms at the Fz sensor. However, this possible effect is masked by noise and is a bit more than speculation with this amount of data; also, the complete lack of this at the Cz sensor is rather worrying, as the ERP components usually are detectable across several adjacent electrodes.

Also, the only reported results are rather late in their latency, regarding the comparable human auditory EEG results: the differences reported are in the time windows of 650-800 ms after the stimulus onset. This is rather worrying, taken that the duration of the auditory stimuli is reported to be, on average, 650ms (p. 13, second row). This raises the question whether the obtained statistical differences are due to the content of the auditory stimuli at all, but actually due to the stimulus offset, perhaps related to the artifacts related to the sound ending (e.g. shifting of position, tension of the ears).

Taken together, I understand the need of all science authors to publish their work, and the setup of this experiment has indeed taken a lot of time and effort. At the same time, the conclusions drawn from the experiments should be always well justified by the measured data, and in this case, I regard the results as unreproducible. Thus, my friendly suggestion for the authors is the following: pool all the data together to obtain more rigorous and general data, and publish your study as an experimental setup for obtaining auditory event-related data from dogs. I think the current conclusions are unreliable, but the work is a valuable contribution to the progress and development of canine neuroscience as a more methodological groundwork.

Appendix B

Response to reviewers

Dear Editor,

We are grateful for the Reviewers' insightful comments and suggestions which we followed carefully during the revision of the manuscript. We would like to summarize shortly the major changes in the revised manuscript compared to the original submission. We believe that these changes resulted in a methodologically stronger, and more convincing paper.

We understood the criticism about the data-analysis, which involved a low number of trials. Therefore, we conducted a second analysis where we changed the data-cleaning pipeline. This resulted in a considerably higher number of cleaned trials compared to the earlier analysis. We conducted the same statistical analyses (using sliding time-windows) as in the earlier version of the manuscript. Importantly, we got similar condition differences in similar time-windows as with our earlier analysis. Additionally, an earlier time-window (between 200-300 ms) also showed significant differences between conditions using the new data cleaning procedure. In the light of the new results, we also updated the abstract and the Results and Discussion. We do not claim anymore that word processing in dogs occurs later than in humans. We report both sets of analyses (analysis of the data cleaned in the original way and analysis of the data cleaned with a new pipeline) in the revised manuscript. We think that the second analysis of the data supports our original results of condition differences and considerably improves the replicability potential of the study.

In order to check whether condition differences are affected only by extreme values of a few dogs, we also tested condition differences with non-parametric tests which are reported in the Results and Discussion and in the Supplemental Results. We also included individual ERPs of each dog in the Supplemental Results, and we extended the description and explanation of our EEG setup including a new figure. We also submit a new Supplementary Material with our manuscript.

Below we reply to each specific comment in turn.

Response to Reviewer 1

Reviewer 1:

General comments:

*The study addresses a theoretically interesting question about phonological perception of whole words in dogs. These findings are likely to be of interest to the general public as well as to developmental psychologists, cognitive neuroscientists, and researchers interested in animal cognition. Non-invasive event-related potentials (ERPs) were used to examine patterns of canine brain activity to perception of mispronunciations of familiar words. Since Pat Kuhl's work seminal work with chinchillas in the 1980s, scientists have known that a variety of non-human animals show robust categorical perception of human speech contrasts. Similar to the present study, non-invasive ERPs with young border collies revealed that dogs perceive human consonant/vowel speech sounds categorically (Adams, Molfese, & Betz, 1987 *Not cited in the current paper). However, this is the first study to examine brain activity to familiar words in dogs. The results suggest that dogs, like 14-month old human infants, treat mispronunciations of known words as acceptable instances of that word. The ERP results are strengthened by consistent findings in a behavioural experiment in dogs' perception of Hungarian words. The findings have implications for launching a plethora of studies examining cognitive development in dogs with implications for human language development.*

Specific comments:

Overall the paper is well-written and a pleasure to read. Figures are very helpful and clearly illustrate the main points.

Author's response:

We thank the Reviewer for the positive evaluation of our manuscript.

Reviewer 1:

Stimuli. The stimuli for the mispronunciations were constructed by changing the vowel (ϵ vs υ for three stimuli; ϵ vs i for one) from a familiar word. In contrast, the nonsense words changed the initial consonant as well as other parts of the consonant/vowel strings. Hungarian has 14 vowels and if I am not mistaken, ϵ vs υ are close in physical distance between formants. This makes the mispronunciations much less perceptible than the nonsense words. A figure or table in supplemental material illustrating physical distance between vowel changes would be helpful. This is not a methodological problem, but it has implications for interpretation of the findings.

Author's response:

The words contained the same sounds across conditions. In this way, we tried to avoid that the newly created words (i.e. the mispronunciations and the nonsense words) would get attention because of different vowels or consonants. Following the request of the Reviewer, we included a Supplementary Material with a figure (Fig. S1) which shows the vowel space (mean F1 and F2 values in our stimuli) of the alternating vowels in the familiar words (WORDS) and mispronunciations (SIMILAR condition). Fig. S1 shows that the F1 values of [ɒ] and [ɛ] are indeed closer to each other, but their F2 values are more different. However, we do not think that difference in F1 and F2 values between vowel-pairs makes the vowel changes in the mispronunciations less perceptible than in the nonsense words. Mispronunciations (words of the SIMILAR condition) contain one word which has the vowel [i] in its first syllable, and likewise, there is only one word with the vowel [i] in the first syllable in the NONSENSE condition. Hence, the distribution of the vowel [i] in the first syllable position is the same for these two conditions. All other vowel changes are between [ɒ] and [ɛ] for the WORDS-SIMILAR and the WORDS-NONSENSE conditions. Therefore, we think that such vowel-changes do not affect the interpretation of the results. In the revised manuscript, we refer to Fig S1 in the Stimulus material section (line 311).

Reviewer 1:

Another difference between the mispronunciations and the nonsense words might be the frequency of the initial phonemes for the nonsense words. They are identical for the known words and mispronunciations. I'm not familiar with Hungarian, but if the initial sounds are less frequent it could account for increased perceptibility between known and nonsense words.

Author's response:

The initial consonants of words are partly different in the NONSENSE condition compared to the words of the other two conditions, because nonsense words were created by mixing the sounds of the WORDS condition. We checked whether there is a difference in the frequency of words starting with the initial consonants across conditions as suggested by the Reviewer. We used the Hungarian National Corpus (MNSZ, http://mnsz.nytud.hu/index_eng.html) which is a database of 187.6 million words of present-day standard Hungarian. The mean frequency of words with the same initial consonants is ranging from 4058.56 to 71792.51 (per million words) for the WORDS and SIMILAR conditions, and from 10953.02 to 71792.51 (per million words) in the NONSENSE condition. The frequencies are in a comparable range, therefore, we think it is unlikely that the frequency of the initial consonants could lead to increased perceptibility between the words of the different conditions. Table S1 contains the frequency values in the Supplementary Material, and we refer to this table in the Stimulus material part of the revised manuscript (line 315).

Reviewer 1:

Another alternative interpretation of the results might be that dogs are sensitive to changes in the initial consonant (as changed in the nonsense words), but not vowels. This is unlikely for several reasons but should be discussed. First, Kuhl showed that in human infants, developmental changes in categorical perception of vowels precedes that of consonants. Second, although 14-month olds did not differentiate between known words and minimal pairs mispronunciations based on changing the initial consonant (Mills et al, 2004), a subsequent study (Mani, Mills, & Plunkett, 2010, DOI:10.1111/j.1467-7687.2011.01092.x) showed that 14-month-olds did show ERP sensitivity to vowel mispronunciations

when a pictorial context was provided. Third, in infants learning Hungarian, phonological perception of vowels occurs quite early in development e.g. see Gonzalez-Gomez, et al. 2019, <https://doi.org/10.1016/j.jecp.2018.08.014>. If dogs, like human infants, rely on distributional learning for phonological cues important for meaning, dogs exposed to Hungarian might be expected to pay attention to vowel changes. A brief but more in depth discussion of physical differences between stimuli is warranted.

Author's response:

We thank the Reviewer for pointing to the physical differences between stimuli. We included a new paragraph in the Results and Discussion (lines 230-240), and in the Stimuli section where we discuss the possible physical differences between stimuli of the different conditions (lines 313-322, 325-328). If we examine the physical differences between stimuli, there are no systematic differences in the mode of articulation between the initial consonants of the different conditions. There is also no systematic difference in the type of vowels of the first syllables across conditions, because the same three vowels were in this position in all conditions. Therefore, it is unlikely that condition differences could arise purely due to sound processing. We are grateful for the references, we included them in the Results and Discussion (lines 253-263) where we discuss whether the vowel manipulation we applied in creating the SIMILAR condition would be perceptually less salient than the consonant manipulation applied by Mills and her colleagues (Mills et al., 2004).

Reviewer 1:

EEG analysis.

Reference. Choice of a reference with dogs must be challenging due to a variety of factors. Using an active electrode site as the reference should have the same associated problems as it does in human EEG research. Other EEG studies with dogs use a non-active common reference, e.g. on the nose. In the present study, ERP amplitudes at Pz would be subtracted from the other active sites, Therefore, it would make sense that ERP differences would be smaller at Cz than Fz because it is closer to Pz. Justification of the choice of Pz as the reference should be included in supplementary materials.

Author's response:

We thank for the opportunity to clarify our choice of reference. This did not get explicitly mentioned in the original manuscript, although we referred to a study (see below) for justifying the electrode-setup. We used an electrode-setup which has been developed and validated in our lab for dogs' sleep EEG [Kis et al. (2014). Development of a non-invasive polysomnography technique for dogs (*Canis familiaris*). *Physiol. Behav.*, 130, 149–156.]. Several studies have already used this setup [e.g. Kis et al. (2017). The interrelated effect of sleep and learning in dogs (*Canis familiaris*); an EEG and behavioural study. *Sci. Rep.*, 7(1); Iotchev et al. (2019). Age-related differences and sexual dimorphism in canine sleep spindles. *Sci. Rep.*, 9(10092)].

In the referred experiments and in our study as well, the dogs were untrained and fully awake during electrode setup, therefore, it would have been problematic to put an electrode on the nose. An earlier study also used a reference different from the nose in awake dogs [Howell et al. (2012). Development of a minimally-invasive protocol for recording mismatch negativity (MMN) in the dog (*Canis familiaris*) using electroencephalography (EEG). *J. Neurosci. Methods*, 89(1),8-13]. We called our reference electrode Pz because it was approximately in the same distance from Cz as Fz but in the posterior direction. However, Pz was placed on a head-surface above the back part of the external sagittal crest (*crista sagittalis externa*) at the occipital bulge of dogs where the skull is usually the thickest and under which either no brain or only the cerebellum is located, depending on the shape of the skull. Therefore, this placement provides a good base for reference as less brain activity can be seen here, and it provides a good control for artifacts as it is close to the other electrodes.

On the other hand, using Pz as a reference could potentially lead to an attenuated effect on Cz compared to Fz. We discuss now the spatial distribution of the ERP effect in the Results and Discussion (lines 225-229), we refer to our reference article of the EEG setup (Kis et al., 2014) and we added a more extended description (lines 355-372) and a figure with the electrode-setup (Fig.5) in the Material and Methods section.

Reviewer 1:

Artefact rejection. The procedure for artefact rejection was clearly presented and the method rigorous. According to Figure 1B & C, 24.87% were clean trials. That would be about 20 trials per condition. Yet, page 6 line 22 indicates there 59 trials per condition “One-way ANOVA showed no differences in the number of clean trials ($M=59$, $sd=10.57$) between conditions ($F(2,32)=1.143$, $p=.167$) (Fig.1C).” This is confusing and should be rewritten to reflect the number of trials per condition.

Author’s response:

We apologize for the misunderstanding. There were 59 trials per dog on average, and there were 18-21 trials per dog per condition on average. We corrected the mentioned sentence in the manuscript (lines 103-105), and now also provide the average number of trials per condition in a table (Table 2 in Methods).

Reviewer 1:

Additionally, the low number of trials per condition could be a concern. There were 80 trials per word type. In human infant ERP research, a recent study showed that when the number of trials per condition is low (i.e. 10-20 trials per condition), measurements and statistical outcomes of the resulting “clean” data can vary depending on the artefact treatment method chosen. The method of using the criteria of a minimum of 10 trials per condition might have been best practice a decade ago, but increasing the number of trials per condition to ensure replicability is of concern. One way to increase the number of trials might be to adjust the high pass filter settings (see next paragraph).

Author’s response:

We experimented on dogs kept as companion animals because they are living in a human-language environment. Therefore, we could not use any anaesthetics or movement restriction during the experiment. Neither did we train our dogs for lying down in a very fixed posture because EEG requires a relaxed position where muscles are not tensed. Hence, our data suffered from movement-artifacts in a degree similar to studies with human infants. Therefore, we followed criteria used in infant EEG studies in the number of trials minimally accepted. Recent infant studies accepted participants’ data with minimum of 10 trials per condition. For example, Sirri et al.’s study [Sirri et al., (2020). Speech Intonation Induces Enhanced Face Perception in Infants. *Scientific reports*, 10(1), 3225)] used a minimum of 10 trials with 17-21 trials per condition, and Forgács et al.’s study [Forgács et al., (2018). Fourteen-month-old infants track the language comprehension of communicative partners, *Developmental Science*. e12751] also accepted infant’s data with minimum 10 trials per condition which resulted in 14-15 trials per condition on average. We were aware that the low number of trials could lead to noisy data, therefore, we also had an additional criterion. The baseline of the subject-averaged data had to be flat with initially aligned ERPs. This criterion was also applied in the mentioned infant studies. Our criteria for inclusion are included in the EEG artifact-rejection and analysis section (Material and Methods, lines 399-403).

However, we agree with the concerns about the low number of trials. Signal to noise ratio of dogs’ EEG might be worse than that of human infants because the dog’s head is covered with fur and more muscles. Therefore, we conducted an additional analysis which resulted in more trials per dog. We explain this in the next response.

Reviewer 1:

Filter settings. The filter settings of .01 to 40Hz are commendable. However, I wonder if a larger number of trials could be included if the off-line high pass filter was set to .1 or even .3 – neither of which should distort the data. More trials per condition would help with signal to noise ratio as well as help increase the potential for replicability.

Author’s response:

We conducted an analysis where filter setting was changed to 0.3 Hz, and the first step of our artifact-rejection pipeline, i.e. the automatic artifact-rejection step was performed. After this step, we got only 27 trials (altogether across all dogs) more compared to the amount of trials after the original automatic

artefact-rejection (with highpass-filter 0.01 Hz). This gave only 1.5 trials more per dog on average (across conditions). Hence, a change in the filter-settings alone did not lead to a considerable increase in the amount of trials.

However, we agree with the Reviewer's concern about the low amount of trials which was also pointed out by Reviewer 2. Therefore, we did an alternative cleaning procedure which resulted in a higher number of trials (see Fig.2 and Table 2). We cleaned the data of the 17 dogs by only applying automatic artifact-rejection (one-step datacleaning). Filter setting followed the Reviewer's suggestion, a 0.3 Hz high-pass and a 40 Hz low-pass filter was applied. Although we used a less rigorous method by leaving out the steps of the video-analysis and the visual inspection of the EEG data from the artifact-rejection, we used stricter thresholds than in the original data-processing pipeline in order to eliminate more movement-artifacts by automatic means.

Trials were cleaned in 100 ms long sliding windows up to 1 s after stimulus presentation. Trials were removed if the amplitudes of the baselined EEG exceeded $\pm 100 \mu\text{V}$ (earlier: $150 \mu\text{V}$) or if the difference of the minimum and maximum values was larger than $120 \mu\text{V}$ (earlier: $150 \mu\text{V}$) in any of the 100 ms time-windows. This procedure resulted in 36-38 trials per condition per dog on average (min=18) (Table 2).

Results of the sliding time-window analyses are now also reported for both the original three-step and the one-step analyses (lines 123-161, Fig.2,3). There are slight differences in the timing of the differences between condition-pairs in the time-window analysis of the data cleaned in one step compared to the analysis of the data cleaned in three steps. Differences between WORDS and NONSENSE were present for a longer duration, between 650 and 800 ms, while ERP differences between SIMILAR and NONSENSE were present between 700-800 ms. There was also an interaction between channels and conditions for this contrast in this time-window. While there was no significant difference between SIMILAR and NONSENSE at Cz, Fz showed a difference. In addition to the results of the original analysis, we also found an effect of conditions between 200-300 ms. Pairwise comparison of conditions showed that ERP was higher for WORDS compared to NONSENSE, and for SIMILAR compared to NONSENSE, and there was no difference between WORDS and SIMILAR. There was also a main effect of channels in this time-window, ERP for WORDS and NONSENSE at Fz was more negative than ERP at Cz. The early time-window (200-300 ms) indicate that the temporal dynamics of word-processing in dogs might be more similar to human processing of words than we have claimed in the earlier version of the manuscript. We included the new analysis and revised the discussion on the timing of the effects in the Results and Discussion part (lines 210-222).

These results also demonstrate that an analysis with considerably more trials also shows similar effects as our initial analysis. However, we think that our initial analysis is also important because it tries to eliminate movement-artifacts in a qualitative analysis by inspection of the video-recordings and the EEG. Therefore, we still include the earlier analysis of the data (three-step data cleaning) in the revised manuscript together with the new analysis containing more trials (one-step data cleaning). We updated the Fig. 2-4 which now show also the results of the new analysis, and we conducted all analyses (e.g. the effect of dog owners' word usage frequency on the average ERP difference of WORDS and NONSENSE conditions at Fz) on the data cleaned in one step as well.

Reviewer 1:

Results.

The time windows were selected based on 100 ms moving windows (with 50 ms overlap) from 0 to 1000ms. This procedure is based on a previous study with dogs, and is a widely accepted method in human ERP studies. In the Mills et al. 2004, human infant study on which the current study is based, ERP amplitude differences were reported between 200 – 500 ms. Because the timing of ERP amplitude differences between dogs and humans is of particular interest, and because the ERP waveforms in the present study appear to show large amplitude differences at Fz from 200-500 ms, it would be helpful to include p values for those windows, even if it is in supplementary material. I'm sure that had those differences been significant, it would have been reported. But it would be nice to get an idea of where those apparent amplitude differences came from in terms of individual variability.

Author's response:

We checked ERP differences between 200-500 ms for the data cleaned in three steps and in one step. There was no effect of condition on the ERPs either in the original data (cleaned in three steps) ($F=1.180$, $p[\text{HF}]=0.3202$) or in the new data (cleaned in one step) ($F=2.337$, $p[\text{HF}]=0.1129$). If we understood correctly, Mills et al. (2004) used a time-window between 200-400 ms, therefore, we also checked whether there were differences in this time-window. The results showed an effect neither in the data cleaned in three steps nor in the data cleaned in one step, although there was a trend for differences in the later one. However, the sliding time-window analysis of the data cleaned in one step revealed differences between 200 and 300 ms, as we discussed in our earlier response. In the Results and Discussion of the revised version of the manuscript, we report the results of the analysis of the time-window between 200-400 ms (lines 116-122) and the results of the sliding-time window analysis for the data cleaned in one and in three steps.

Regarding individual differences, we also included figures with individual ERP results of each dog in the Supplemental Results (Fig.S2). In order to judge whether differences between conditions are led only by a few individual dogs or whether most of the dogs show differences between conditions in the same direction as the significant effects, we also conducted an additional analysis, a Wilcoxon-signed rank test in the selected time-windows. The results of this analysis showed that most of the dogs showed an effect in the direction which was significant by the ANOVA-analysis in most of the selected time-windows (except of one where p was 0.051). We also included the results of this analysis in the revised manuscript (lines 165-168 in Results and Discussion, Table S2 in Supplemental Results).

Reviewer 1:

Discussion

Page 12 lines 15.16: "However, we also show here that important aspects of word processing are not shared between the two species. The ERP findings suggest that dogs' word processing capacities are relatively slow and sensitive to the frequency with which dogs encounter the words."

Actually, sensitivity to phonological and word frequency is something dogs have in common with humans across development. This has been shown in a large number of ERP studies as well as other behavioural studies. See papers below (just suggested not necessary to include these references – but frequency affects phonological and word processing in humans across the lifespan). Indeed, this is one of the main strengths of the paper! Figure 3 showing individual differences illustrating the ERP Word – nonsense ERP amplitudes is very impressive. Showing that these data replicate results from studies with human infants enhances the impact and believability of the study.

An important difference in phonological perception between dogs and human infants is likely to be on attention allocation. Early in development, human infants rely on domain general processes to pick up on statistical regularities (e.g. transitional probabilities within versus between words) in the speech stream to segment words from continuous speech. With increased experience infants learn to use more domain-specific information such as stress and prosodic cues such as infant-directed speech. On the other hand, dogs are more likely to learn isolated words as commands associated with food or social rewards. What is so remarkable about the present study are the similarities, not the differences, between dogs and human infants in the way the brain processes familiar words.

Phonological development:

*Sita Minke ter Haar & Clara Cecilia Levelt (2018) Disentangling Attention for Frequency and Phonological Markedness in 9- and 12-Month-Old Infants, *Language Learning and Development*, 14:4, 279-296, DOI: 10.1080/15475441.2018.1480375*

*Swingley D. (2009). Contributions of infant word learning to language development. *Philosophical transactions of the Royal Society of London. Series B, Biological sciences*, 364(1536), 3617–3632. doi:10.1098/rstb.2009.0107*

Experience with individual words:

*Mills, D. L., Plunkett, K., Prat, C., & Schafer, G. (2005). Watching the infant brain learn words: Effects of language and experience. *Cognitive Development*, 20, 19-31.*

Author's response:

We are grateful for the insightful comments on our conclusion. We revised the discussion of the results. In the light of the new results (i.e. the early effect between 200-300 ms in our second analysis of the data cleaned in one step), we note that the temporal dynamics of dogs' word processing might be similar to human word processing (lines 217-220). In the last paragraph of Results and Discussion we write that the frequency effect and the WORDS-NONSENSE differences are in line with earlier results showing analogies of word processing between dogs and humans (line 277-278).

In the first version of the manuscript, we emphasized that similarities in processing instruction words and their mispronounced version might be due to weaker word representations because weaker word representations could also explain well the late effect of word processing. However, we also mentioned (but not emphasized) already in the earlier version of the manuscript that attentional demand can also be a potential factor behind the lack of discrimination of similar words. We agree with the reviewer that attention allocation might be a difference between humans and dogs, and it seems to be a more likely explanation in the light of the new results. In the revised manuscript we left out a paragraph discussing weaker word representations and we conclude at the end of Results and Discussion that dogs might differ in allocation of attention from humans (lines 279-281). We are grateful for the suggested references, we included them in the Results and Discussion (lines 241-243).

Reviewer 1:*Minor comments:*

Add reference below to sentence: "Fully non-invasive EEG has been applied in earlier studies on awake but trained dogs (17,18)."

*Christina L. Adams, Dennis L. Molfese & Jacqueline C. Betz (1987) Electrophysiological correlates of categorical speech perception for voicing contrasts in dogs, *Developmental Neuropsychology*, 3:3-4, 175-189, DOI: 10.1080/87565648709540375*

Author's response:

We are grateful for the suggested reference. In the revised manuscript, we refer to the study by Adams et al. (1987) (in lines 40 and 66), but we inserted it in a new sentence (line 66) because dogs were tranquilized in their study, hence, the method of the study was invasive.

Response to Reviewer 2

Reviewer 2:

The article by Magyari and colleagues is of a highly timely topic and the authors, together with possible research assistants, have obviously put considerable amounts of work and effort into this manuscript. Personally, I would love to see more research on this topic. However, the first studies of this kind need to be absolutely trustworthy, since all the forthcoming related science will rely on them. I am sad to say that this manuscript does not meet the quality standards that we need for this kind of work studying neuroscientific basis of dog cognition.

My main concern in this paper is the sufficiency and the quality of the data that all the discussion is based on. I am seriously concerned that the results reported in the paper are due to false positive findings, and they are not reproducible, for the following reasons.

The authors report that they have obtained approximately 20 clean event-related responses of each of the three auditory conditions per dog. They state "mean number of clean trials (y-axis) per condition (x-axis)" on the Fig. 1 legend (page 8/lines 93-94), and the Figure 1 y-axis values approximate +/- 20. Taken the signal-to-noise ratio of electroencephalography into account, obtaining a difference of event-related potentials to 20+20+20 stimuli is highly unreliable also in human EEG data, and the SNR of the dog EEG data is far worse due to the signal decay over distance and the conductance differences in the tissues between the signal source and the measurement site. Also, the impedances of dog EEG tend to be worse than those of human EEG, lowering the SNR. Also in this study, the impedances are

reported to be 15 or lower. Thus, the event-related samples of approx. 20 trials per dog per condition, with $n=17$ participant dogs' data included, are inadequate to reach any final conclusion.

Author's response:

When we started this project, our main concern with earlier ERP studies of dogs was that those applied no artifact-rejection based on qualitative data-analysis (e.g. visual inspection of EEG data or video-analysis for movements). And we also know that it is difficult to ensure that dogs would not move or tense their muscles without any invasive method. Therefore, using only automatic artifact-rejection methods seemed to be insufficient for data-analysis. Hence, we used a rigorous method for artifact-rejection. The pipeline for this process was based on studies of infant research where movement-artifacts are a similarly big concern. We followed criteria used in infant EEG studies in the number of trials minimally accepted. Recent infant studies accepted participants' data with a minimum of 10 trials per condition. For example, Sirri et al.'s study [Sirri et al., (2020). Speech Intonation Induces Enhanced Face Perception in Infants. *Scientific reports*, 10(1), 3225] used a minimum of 10 trials with 17-21 trials per condition, and Forgács et al.'s study [Forgács et al., (2018). Fourteen-month-old infants track the language comprehension of communicative partners, *Developmental Science*. e12751] also accepted infant's data with minimum 10 trials per condition, which resulted in 14-15 trials per condition on average. We were aware that the low number of trials could lead to noisy data, therefore, we also had an additional criterion. The baseline of the subject-averaged data had to be flat with initially aligned ERPs. This criterion was also applied in the mentioned infant studies. Our criteria for inclusion are included in the EEG artifact-rejection and analysis section (Material and Methods, lines 399-403).

However, we understand the Reviewer's concern about the number of trials and the replicability of the results, especially because dogs' EEG may be worse in signal-to-noise ratio than human infants' EEG. Therefore, we performed an additional analysis, where we used only automatic artifact-rejection with a higher high-pass filter (0.3) and with stricter thresholds compared to the first step of our original artifact-rejection pipeline. Trials were removed if the amplitudes exceeded $\pm 100 \mu\text{V}$ (earlier: $150 \mu\text{V}$) or if the difference of the minimum and maximum values was larger than $120 \mu\text{V}$ (earlier: $150 \mu\text{V}$). This method resulted in more trials 36-38 in average per dog per condition (see Fig.2 and Table 2 in the revised manuscript). This analysis showed a similar pattern of differences between conditions and slight differences in the timing of the condition differences (Fig.2,3). Differences between WORDS and NONSENSE were present in a longer duration, between 650 and 800 ms while ERP differences between SIMILAR and NONSENSE was present between 700-800 ms. There was also an interaction between channels and conditions for this contrast in this time-window. While there was no significant difference between SIMILAR and NONSENSE at Cz, Fz showed a difference. In addition to the results of the original analysis, we also found an effect of conditions between 200-300 ms. Pairwise comparison of conditions showed that ERP was higher for WORDS compared to NONSENSE, and for SIMILAR compared to NONSENSE and there was no difference between WORDS and SIMILAR. There was also a main effect of channels in this time-window, ERP at Fz was more negative than ERP at Cz. This analysis is also included in the revised manuscript (lines 120-122, 149-161).

Condition differences of the second analysis involving a higher number of trials showed the same pattern as our earlier analysis, and a later time-window of differences was also present in the second analysis. Therefore, we think our original results are likely not due to false positives. On the contrary, it seems that the original data cleaning pipeline is more rigorous which might lead to elimination of experimental effects. However, further research should confirm the robustness of the early effect (between 200-300 ms) revealed only in our second data cleaning procedure. Therefore, we report the results of both analyses in the revised manuscript.

Impedances were reported to be 15 or lower. Actually, we aimed for keeping impedances under 5, especially for Fz and Cz. In fact, we have lab notes about actual impedances for 10 out of 17 dogs. For 8 of these 10 dogs, impedances were under 5 at Fz and Cz. However, for a few dogs it was sometimes impossible to reach low impedance for electrodes around the eyes (F7 and F8). Even in those cases when impedance values of F7 and F8 were just below 15, we only started the experiment when correlates of eye-movements (e.g. blinks or eye-brow movements) were visible on F7 and F8.

Reviewer 2:

Furthermore, there is no mention of background noise conditions in the room where the recordings take place. Was the recording room properly soundproof, as this is an auditory study? Were there any ambient background noise in the room? Was anyone else present than the researcher, owner and the dog? Did the dog stay still during the measurements?

Author's response:

When we wrote the paper we aimed for a short format, and we realize now that some information was not reported in the Methods section. We apologize for the mistake.

The recording took place in a dedicated, windowless room which is in the basement of the university. Only those entered that part of the corridor who are coming for experiments. There are also another experimental rooms in the same corridor but we took care to schedule our experiments when other experiments were not conducted. The room was not soundproof. Soundproofing would have certainly improved our chances for better SNRs but, importantly, the only ambient environmental noise was continuous and soft (air conditioner). Also, multiple persons needed to be present in the same room, thus having a fully silent setting was not an option anyways (this is similar to recording with infants). Two experimenters and the owner were present in the room. If the dog was accompanied by two owners, we asked one of them to return later when we finished with the experiment. We also wrote in the Procedure section that owners were asked to sit on a mat with their dog next to them as if it was a relaxation period for them and for their dog. This relaxation happened most of the time by lying down, and in some rare cases by sitting. When the experiment started, dogs usually were also putting their head down on the ground after the first few trials. If dogs stood up during the experiment, we asked the owner to try to put the dog back to lying position. If the dog wanted to leave from the mat, the experiment was aborted.

We updated the Procedure section with more information (lines 336-346).

Reviewer 2:

The placing of electrodes would also need more justification, as there are no common standards yet in the dog neuroscientific research. Why did the authors decide to measure the data from these locations? How did the authors decide the location of the reference electrode? Why this number of electrodes? And finally, how did the authors take the previous dog ERP research into consideration when deciding these?

Author's response:

We apologize that our choice for electrode-setup did not get explicitly mentioned in the original manuscript. We used an electrode-setup which has been developed and validated in our lab [Kis *et al.* 2014. Development of a non-invasive polysomnography technique for dogs (*Canis familiaris*). *Physiol. Behav.* 130, 149–156.] and several studies have already used [e.g. Kis *et al.* 2017. The interrelated effect of sleep and learning in dogs (*Canis familiaris*); an EEG and behavioural study. *Sci. Rep.*, 7(1); Iotchev *et al.* 2019. Age-related differences and sexual dimorphism in canine sleep spindles. *Sci. Rep.*, 9(10092)]. Although this setup has been used for sleep research in dogs, we aimed to use the same setup for awake ERP. In this way, we can connect results of awake cognition to sleep in future research. We referred to our reference article of the EEG setup [Kis *et al.*, 2014] in the Material and Methods section in the earlier version of the manuscript, but we have not explicitly mentioned it. We now included an extended explanation and a new figure (Fig.5) about the electrode setup in the Material and Methods section (lines 354-372).

Reviewer 2:

Images of event-related responses on individual dogs would be also very informative, and the numbers of trials that reached analysis from each of the participant dogs. From the current data, it is impossible to determine e.g. if the Grand Average response depicted in Fig. 2A is mainly due to a few dogs' leading effect, or whether the GA indeed represents a valuable average of the whole population.

Author's response:

We included the individual ERP figures of each dog's data in the Supplemental Results. The figures also present number of trials in each condition (Fig. S2). We also conducted Wilcoxon-signed rank tests (see Table S2 in Supplemental Results) in the selected time-windows. This non-parametric test ensures that condition differences are not only due to some extreme values of individual dogs' data. The number of dogs having condition differences in the same direction as condition differences of the grand-average are also reported in Table S2 for each time-window. In each time-window 11-13 dogs showed condition differences in similar direction out of 17. Condition differences were significant in all relevant time-windows for the WORDS-NONSENSE contrast for both analyses, and also for the SIMILAR-NONSENSE contrast in the time-windows following one-step-cleaning. The only non-significant one of all tested contrasts was for the SIMILAR-NONSENSE contrast following three-step-cleaning where p was 0.051. Therefore, it is likely that results are not due to the leading effect of a few dogs. We report the results of these non-parametric tests in lines 165-169.

Reviewer 2:

In addition, standard error levels should be added to the Fig 2A, just to clarify how much of the response can be distinguished from the noise levels.

Author's response:

We included standard error shades on the ERP grand-averages on Fig.2 in the revised manuscript.

Reviewer 2:

Looking at the Fig. 2A as it is now, one could deduce that there is a possibility of an effect of the words vs. nonsense conditions around 200ms at the Fz sensor. However, this possible effect is masked by noise and is a bit more than speculation with this amount of data; also, the complete lack of this at the Cz sensor is rather worrying, as the ERP components usually are detectable across several adjacent electrodes.

Author's response:

In our second analysis, where we used only automatic artifact-rejection, we found ERP differences between 200-300 ms. We updated our conclusion accordingly. We do not state now that dog's word processing would have a later timing compared to humans, instead we note that the temporal dynamics of dogs' word processing might be comparable to that of human word processing (lines 210-222).

The size of the head of dogs varied, therefore it is difficult to judge whether any effect at Fz should be also present at Cz due to volume conduction. We think that the attenuated effects at Cz could be caused by two factors: Either the results reflect a frontal effect, or the attenuation is simply caused by the proximity of the reference electrode, Pz, which is closer to Cz than to Fz. We discuss this in the Results and Discussion in lines 225-229.

Reviewer 2

Also, the only reported results are rather late in their latency, regarding the comparable human auditory EEG results: the differences reported are in the time windows of 650-800 ms after the stimulus onset. This is rather worrying, taken that the duration of the auditory stimuli is reported to be, on average, 650ms (p. 13, second row). This raises the question whether the obtained statistical differences are due to the content of the auditory stimuli at all, but actually due to the stimulus offset, perhaps related to the artifacts related to the sound ending (e.g. shifting of position, tension of the ears).

Author's response:

In our second analysis of the data using only automatic artifact-rejection, there were also ERP effects between 200-300 ms. Condition differences showed the same pattern as in our original analysis in the late time-windows. Therefore, we think that similar processing might occur at the early and the late

time-windows. Even if there was an effect of sound endings or muscle tension undetectable by inspection of the EEG and the video-recordings, we think they could not cause differences between conditions as there was no significant difference in the duration of the sound stimuli across conditions (see Methods, lines 326-328).

Appendix C

The manuscript by Magyari and colleagues represent a revised version of the previous submission (as a new submission).

The authors respond very nicely to some of my previous concerns, for the previous submission, about measurement settings, choice of electrode placements, length of the stimulus conditions etc., which are now waived. However, my major concerns about the data sufficiency and the quality of the data still remain, and I see my previous suggestion of reporting this study as a more methodological paper on dog auditory processing was simply omitted from the letter. (I quote myself: “Thus, my friendly suggestion for the authors is the following: pool all the data together to obtain more rigorous and general data, and publish your study as an experimental setup for obtaining auditory event-related data from dogs. I think the current conclusions are unreliable, but the work is a valuable contribution to the progress and development of canine neuroscience as a more methodological groundwork.”)

The main point of the review process is to use the Occam’s razor and try to ask whether there are any simpler explanations for the obtained results, and in my opinion, there still is.

The authors now report that they have conducted “a second analysis where we changed the data-cleaning pipeline.” With the new automatic data preprocessing, 36-38 trials per condition per dog was acquired. In practice, it means that the authors have allowed more data in the analysis, but the data is still coming from the same measurement and is likely containing some artifacts, since it was removed within the previous preprocessing steps.

Number of trials per condition remains a potential confound factor, the second “one-step automatic artifact-rejection”, allowing previously rejected trials into averages, does not improve the situation (it reminds a “double dipping” procedure, the same data cannot confirm itself). I agree with the authors that visual inspection of the data is needed, this was not one of my concerns. The supplementary material of individual dog responses are extremely helpful in understanding the data, but looking at them together with Fig. 2, one cannot get a clear picture of the general response form. Even the auditory N1 responses are not detectable, and this is an auditory study where, despite the location of the electrodes, we should see at least a hint of the primary auditory response (in Howell et al 2011, the first auditory response is detectable in a similarly-placed Cz electrode, even if it is away from the auditory cortices). I would probably have tried a bit lower LP filter to get rid of the obvious high-frequency noise, but I don’t know if it would help here.

In their letter, the authors refer to the infant studies, which methodologically provide a similar framework than measuring EEG from dogs (i.e. the subject can be restless and moving), and justify their choices by the number of trials accepted in infant studies. However, human infants have very thin skull & skin and virtually no musculature on top of the brain, their heads are even “better” for measuring EEG on this behalf than those of adults. Instead, compared to adult humans, dogs are exactly the opposite. They have much thicker musculature and thick skulls covering the brains, topped with fur. Therefore, from infants, one may be able to record clear responses with few trials – maybe even fewer than in human adults – but from dogs, one should be able to get more trials than standardly from human adults to reach a similar sensitivity.

There also is a big possibility for behavioral effects in the late-latency ERP differences, between 650 and 800 ms. It is possible, even likely, that changes measured in the electrical activity with this late latency are caused by behavioral effects instead of neural events.

Finally, I repeat that I think this study is a valuable contribution to the literature on a larger scale, but I am still not convinced that the category-dependent results discussed here are due to true effects. I am also puzzled by that the authors do not accept even the possibility of the potential confounds, when it should be also their own duty as scientists to look for simplest possible explanations.

Appendix D

Response to Editor and Reviewers

Editor:

Associate Editor Comments to Author (Dr César Lima):

We invite the authors to address the remaining concerns in a major revision, paying particular attention to the power issues raised by Reviewer 2. This shortcoming should be explicitly acknowledged and discussed, and the N should be included in the abstract.

As noted by Reviewer 1, inferences based on indirect comparisons (between dogs and humans) also need to be made more cautiously, and clearly phrased as speculative and for future work. The same applies to inferences based on null results, as these do not provide evidence for the null hypothesis (e.g., processing similarities between known words and similar nonsense words). In fact, running Bayesian analyses would be ideal if the authors want to formally interpret these findings.

Finally, the authors should also expand on the methodological contribution of their work.

Dear Editor,

We are grateful for the possibility to address the remaining concerns in a major revision.

We addressed the power issues raised by Reviewer 2. In the Results and Discussion, we discuss the low number of participant and trials, and we note that our second analysis (one step-analysis) cannot confirm the results of the first analysis as both analyses were conducted on the same data (lines 228-248). We also report the number of participants in the abstract (line 26). We also toned down our claims at several places in the text by referring to the need for future studies (abstract: lines 34-35, Results and Discussion: lines 247-248, 313, 316, 319-320, 322, 324) and by the indication of speculation (line 310, 321). We address more cautiously the comparison of our results to results of human ERP studies (line 310). We also left out the sentences “We propose that the temporal dynamics of word processing in dogs might be comparable to that in humans...” from the abstract and “However, our findings also suggest that dogs are less efficient in discriminating words based on phonetic details compared to capacities of human adults” from the last paragraph of the Results and Discussion section. We emphasize more the methodological contribution of our study in the abstract (lines 25-29) and in the Results and Discussion (lines 92-93, 95-96, 234-236, 314-315).

Further changes relate to the specific requests of the reviewers: We discuss the possibility of behavioural effects in the late time-window (lines 255-262), the differences between the effects in the early and late time-windows (lines 281-283), the criteria for accepting dogs’ data with a minimum of 10 trials per condition (lines 440-441), and the general shape of the ERPs (Supplementary Material, Supplemental Results and Discussion). Following the suggestion of Reviewer 2, we also included a figure in the Supplementary Material with the ERP responses after a one-step automatic cleaning procedure in which the low-pass filter was changed to 20 Hz. We refer to this part of the Supplementary Material in lines 121-125 of the Results and Discussion.

Following the instructions in the latest email, we rearranged the following sections: Ethics statement, Data accessibility, Competing interests, Author’s contributions, Acknowledgement and Funding statement. These are placed before the reference list in the revised manuscript.

We submit two copies of the revised manuscript. In of them, changes are highlighted.

Below we reply to each specific comment in turn.

Response to Reviewer 1

Reviewer 1:

Comments to the Author(s)

The authors are to be commended for their careful response to methodological concerns regarding the stimuli and data analysis. As this study is one of the first of its kind and might be used as the methodological standard for subsequent research. Thus, it is important to stress methodological rigour. The frequency table for the initial phonemes and addressing the possibility of physical differences between the mispronunciations and nonsense words in the discussion were helpful. Examination of the effects of different approaches to artefact treatment on statistical analyses across different time windows, has important methodological implications for EEG research with human infants as well. As a reader who is interested in EEG methodology, I found this quite engaging. However, I wonder if the typical RSOS reader would find the article easier to process if only the one -step analysis was presented – and the comparisons of different artefact treatments were presented in the supplemental materials.

Author's response:

We understand that the article might be a bit too difficult to read with the two kinds of artifact-rejection procedures. However, Reviewer 2 suggested to report our study as a more methodological paper and the Editor also asked us to expand on the methodological contribution of our work. Therefore, we left both types of analyses in the main text. We also think that artifact-rejection is a crucial part of developing ERP methodology for dogs.

Reviewer 1:

My only other comments are in regard to the conclusion that dogs show limited phonological processing. For example, a) “the temporal dynamics of word processing in dogs might be comparable to that in humans but dog’s capacity to access phonological detail is more limited” (abstract), Also, b) “These findings suggest that reduced readiness to process phonetic details may be one reason that incapacitates dogs from acquiring a sizeable vocabulary” (conclusion). I agree those statements are both likely to be true, but the authors didn’t demonstrate how dogs differed from human infants based on their data. Unless they can explicitly state how the ERP data show limited access to phonological detail, those ideas are speculative and for future research.

Author's response:

In the revised manuscript, we toned down these claims and emphasize more the uncertainty and speculations in our conclusion. We revised the abstract (lines 34-35) and the Results and Discussion (lines 247-248, 313, 316, 319-320, 322, 324). In the last paragraph of Results and Discussion (lines 320-321), the lack of differences is reported and we clearly state that our interpretation about the lack of these differences is speculative (see also line 310).

Reviewer 1:

Of particular interest was the correlation between the owner's word use frequency and the size of the ERP amplitude difference in the late but not early time window. It is possible, that like human infants these time windows reflect qualitatively different aspects of word processing in dogs. Although, as the statistical significance ERP differences in this time window differed depending on the artefact rejection method, it is commendable not to be too speculative. Overall, this approach leads the way to examine many different theoretical questions on non-human sensitivity to the phonology of human speech.

Author's response:

We thank the reviewer for pointing to the issue of the processes underlying ERP differences in the two time-windows. We did not discuss this in the earlier version of the manuscript to avoid speculation. In the revised version, we formulated this issue as an open question (lines 281-283).

Response to Reviewer 2**Reviewer 2:**

The manuscript by Magyari and colleagues represent a revised version of the previous submission (as a new submission).

The authors respond very nicely to some of my previous concerns, for the previous submission, about measurement settings, choice of electrode placements, length of the stimulus conditions etc., which are now waived. However, my major concerns about the data sufficiency and the quality of the data still remain, and I see my previous suggestion of reporting this study as a more methodological paper on dog auditory processing was simply omitted from the letter. (I quote myself: "Thus, my friendly suggestion for the authors is the following: pool all the data together to obtain more rigorous and general data, and publish your study as an experimental setup for obtaining auditory event-related data from dogs. I think the current conclusions are unreliable, but the work is a valuable contribution to the progress and development of canine neuroscience as a more methodological groundwork.")

Author's response:

We apologize for the mistake of omitting the last paragraph of the review from our letter. It was not intentional. The last paragraph got to a second page in the pdf in which we downloaded the review and although we read this paragraph too, we did not include it in the letter later through an oversight. Nonetheless, we intended to revise our manuscript to be more methodological. This is why we included the two artifact-rejection procedures in the main text, and extended the method section. We are aware that this was not exactly what Reviewer 2 suggested, and the reason for that was that we attempted to find a fair balance between the requests of both reviewers.

In the now revised version of the manuscript we emphasize more the methodological aim of the study (abstract: lines 25-29, Results and Discussion: lines 92-93, lines 95-96, 234-

236, 314-315) and we toned down our conclusions in the abstract and in the Results and Discussion (abstract: lines 34-35, Results and Discussion: lines 247-248, 313, 316, 319-320, 322, 324).

Reviewer 2:

The main point of the review process is to use the Occam's razor and try to ask whether there are any simpler explanations for the obtained results, and in my opinion, there still is.

The authors now report that they have conducted "a second analysis where we changed the data-cleaning pipeline." With the new automatic data preprocessing, 36-38 trials per condition per dog was acquired. In practice, it means that the authors have allowed more data in the analysis, but the data is still coming from the same measurement and is likely containing some artifacts, since it was removed within the previous preprocessing steps.

Number of trials per condition remains a potential confound factor, the second "one-step automatic artifact-rejection", allowing previously rejected trials into averages, does not improve the situation (it reminds a "double dipping" procedure, the same data cannot confirm itself).

Author's response:

We agree with the Reviewer that the second (one-step) analysis does not confirm the results as a new experiment would do. However, we think that the results of the second analysis make some of the concerns with the first analysis less likely.

Our first analysis was very conservative, therefore, it resulted in a rather low number of trials (about which both Reviewers were also concerned). When trial numbers are low, random noise or artifacts present only in a few trials can lead to condition differences. Given the rigorous artifact-cleaning procedure, the possibility for trials containing frequent, typical movement artifacts time-locked to the stimulus is very low. However, other types of artifacts, e.g. less frequent and less typical artifacts or random noise can be still present in the cleaned data and could lead to ERP differences when trial numbers were low. However, these types of artifacts might cancel out when more trials are averaged because these are less frequent and/or not time-locked to the stimulation. Hence, the second analysis with more trials and with similar effects makes it less likely that such artifacts could cause ERP effects in the first analysis. It is also very unlikely that less frequent, atypical artifacts cause condition differences in the first analysis, and more frequent, condition-dependent movement artifacts cause a difference in almost the same time-window and between the same conditions in the second analysis. Moreover, if there are many condition-dependent movement artifacts in our data, we could also expect that the number of rejected trials are different across conditions (i.e. some conditions should contain more artifacts), but there were no differences in the number of rejected trials.

We also think that the extra trials present in the second analysis do not necessarily contain more artifacts because the cleaning procedure of the first analysis was very conservative. We rejected all trials with movements visible in the video-analysis, and from the remaining trials, we also rejected trials where the EOG channels (F7, F8 and their bipolar derivation) showed characteristics of blinks and horizontal eye-movements and/or a correlation with Fz and Cz channels. This latest criteria (correlation with Fz and Cz) is motivated by the assumption that any artefact from eye-movements showing up on the Fz and Cz channels will be also present on F7 and F8 because those are more sensitive for eye-movements. But task-

dependent, cognitive effects can also show up on all channels, including F7 and F8. Hence, we think it is also likely that the first analysis was too conservative and clean trials without any artefacts got also rejected. Hence, similar effects could show up in the second (one-step) analysis when more trials were included because we eliminated some of the trials containing the experimental effect in the first analysis. Of course, our interpretation about the two cleaning procedure is speculative and based on assumptions. But we would like to emphasize that the extra trials of the second (one-step) analysis do not necessarily contain artifacts just because they were removed in the first (three-step) analysis. We also would like to point out that the automatic artifact-rejection pipeline of the one-step analysis was run with stricter thresholds than the artifact-rejection pipeline of the three-step analysis. Therefore, extra trials of the one-step analysis are not all of those trials which were rejected by video-inspection and visual inspection of the EEG wave in the three-step analysis.

Artifact-cleaning is a crucial part of ERP data-analysis and there is no gold standard for it. Hence, we think it is a valuable methodological contribution to explore the different artifact-rejection methods. The second analysis also shows that the results are not too sensitive to the method for cleaning, because condition differences found in the first (three-step) analysis also showed up in the second analysis.

In the revised manuscript, we noted now that the second analysis cannot confirm the results of the first analysis but it provides a methodological insight (lines 232-236). We also discuss whether the extra trials of the second analysis contain artifacts (lines 236-248).

Reviewer 2:

I agree with the authors that visual inspection of the data is needed, this was not one of my concerns. The supplementary material of individual dog responses are extremely helpful in understanding the data, but looking at them together with Fig. 2, one cannot get a clear picture of the general response form. Even the auditory N1 responses are not detectable, and this is an auditory study where, despite the location of the electrodes, we should see at least a hint of the primary auditory response (in Howell et al 2011, the first auditory response is detectable in a similarly-placed Cz electrode, even if it is away from the auditory cortices). I would probably have tried a bit lower LP filter to get rid of the obvious high-frequency noise, but I don't know if it would help here.

Author's response:

If we look for a general response form, the visually most observable characteristic of the grand-average is a small negative peak around 100 ms and a larger negative peak between 200 and 300 ms at all conditions in the data cleaned in one-step at the Fz channel. In the individual ERPs, we can see a negativity in most of the conditions for example, at dog 1, dog 6, dog 8, dog 9, dog 11 and dog 12. Therefore, we think that there is a general response form, although it can be seen better in the results of the one-step cleaning (probably due to a better signal-to-noise ratio). In our study, these peaks are more observable at Fz, and less shown at Cz, which is probably due to proximity of the Cz to the reference electrode.

We agree with the reviewer that it would be indeed reassuring if an auditory N1 could be also seen in the data. Although Howell and colleagues (Howell et al., 2012) showed auditory N1-like responses for dogs, they used a reference which was more distant from Cz and they measured EEG with needles inserted under the skin, hence, they probably achieved a better sensitivity for the EEG components. But even if their EEG measurement was more sensitive,

the N1-like response had a relatively small amplitude when six dogs' ERP results were averaged (see Fig.2 in Howell et al., 2012). The N1 had a larger amplitude only when the individual ERP results of two selected dogs were presented (see Fig.3 in Howell et al., 2012). However, we have to note that in another study by Howell et al. (2011), there was no clear N1 response in the ERP of that one dog which participated in that auditory study. In our study, future research could address whether the observed small negativity around 100 ms is similar to an N1 ERP component exhibited by humans. Our experiment was not designed for revealing the N1 component (i.e. we did not expect any modulation of this component), hence, we cannot make any conclusion about it. Future studies could also examine whether the negativity around 200-300 ms is stimulus-specific or also reflects a more general auditory response.

We included a figure (Fig.S2) about the ERPs after a one-step data-cleaning procedure with 20 Hz low-pass filter as suggested by the Reviewer in the Supplementary Material (Supplemental Results and Discussion). Here, we also discuss the shape of the ERPs.

In their letter, the authors refer to the infant studies, which methodologically provide a similar framework than measuring EEG from dogs (i.e. the subject can be restless and moving), and justify their choices by the number of trials accepted in infant studies. However, human infants have very thin skull & skin and virtually no musculature on top of the brain, their heads are even "better" for measuring EEG on this behalf than those of adults. Instead, compared to adult humans, dogs are exactly the opposite. They have much thicker musculature and thick skulls covering the brains, topped with fur. Therefore, from infants, one may be able to record clear responses with few trials – maybe even fewer than in human adults – but from dogs, one should be able to get more trials than standardly from human adults to reach a similar sensitivity.

We thank the Reviewer for pointing out that the ERPs of dogs and infants might be very different with regard to their volume-conduction properties. In the earlier version of the manuscript, we were more concerned about the movement-artifacts which are very high for both dogs and infants. Therefore, we included in lines 440-441 of the revised manuscript where the criteria of minimum 10 trials are mentioned that the number of 10 trials might be too low because dogs have a much thicker layer of muscles, thicker skull and fur on the top of their head.

There also is a big possibility for behavioral effects in the late-latency ERP differences, between 650 and 800 ms. It is possible, even likely, that changes measured in the electrical activity with this late latency are caused by behavioral effects instead of neural events.

Finally, I repeat that I think this study is a valuable contribution to the literature on a larger scale, but I am still not convinced that the category-dependent results discussed here are due to true effects. I am also puzzled by that the authors do not accept even the possibility of the potential confounds, when it should be also their own duty as scientists to look for simplest possible explanations.

We thank the Reviewer for the positive evaluation of our study as a valuable contribution to the literature. We are also very aware of our duty of looking for simplest possible explanations.

Following a rigorous scientific approach, we developed a quite time-consuming artifact-cleaning procedure to make it very unlikely that condition differences are due to condition-dependent muscle movements. Therefore, we think that behavioural confounds might be the simplest but not the most likely explanation. We also would like to note that even if ERP effects are due to muscle movements, those would still show that dogs differentiate word-categories, hence, such a confound would not undermine the claim about dogs' ability for discriminating known words from nonsense words.

Nonetheless, we agree with the Reviewer that behavioural effects can still influence the results even if we tried our best to eliminate those. In the revised manuscript, we discuss the possibility of category-dependent muscle movements for the late effect in lines 255-262.

Appendix E

Dear Editor,

We thank for the opportunity for addressing suggestions and concerns raised by the reviewers in a revision. Here, we report on the major changes committed on the manuscript following the reviewers' suggestions.

In the earlier versions of the manuscript, we applied two methods for cleaning the electrophysiological data. We called these procedures as one-step and three-step cleaning. In the recent version, we renamed these procedures throughout the text to amplitude-based and multi-level artifact rejection. We moved one figure (Fig.S2 in the earlier and Fig.2. in the recent version) and its discussion from the Supplementary material to the main text. We removed one analysis (Wilcoxon-signed rank tests) to the Supplementary material. This analysis was additional statistics included in the main text of the manuscript at the first round of revision, it does not contribute any new information to the main results of the study. We also tested by Bayesian methods the lack of condition differences in the ERP amplitudes for those conditions which were not significantly different by repeated-measures ANOVA (WORDS and SIMILAR conditions) in those time-windows where other pairs of conditions showed a difference. We tested the strength of evidence for the null hypothesis, i.e. the strength of evidence for no differences between conditions. Almost all of the tests showed a substantial evidence for no differences between WORDS and SIMILAR conditions. These results support the conclusion of the earlier version of the manuscript. Therefore, we did not change the conclusion in the recent version. We only changed a phrase from “we speculate” to “we propose” in the last paragraph of the Results and Discussion (line 358).

We submit two copies of the revised manuscript. In one of them, changes are highlighted.

Below we reply to each specific comment in turn.

Response to Reviewer 2

Reviewer 2:

Comments to the Author(s)

Magyari et al, RSOS-200851.R1 "Event-related potentials reveal limited readiness to access phonetic details during word processing in dogs"

The revised work by Magyari et al has now improved with the more cautious interpretation of the data, including the grand-average across conditions situations adding to the replicability of the study and by considering the possible artifacts more carefully. I am glad that the authors have now considered their contribution on a grander scale for the field, how their input piece fits into the big picture. I think this will be an informative contribution to this growing field. I have now gone through the manuscript now in more detail, and I have the following further suggestions or concerns:

Introduction, lines 65-72: “Here we study... - ...method (needles inserted under the skin) (20).”

-> Continuous EEG of dogs have been abundantly measured in the context of disorders such as epilepsy, which could be mentioned. Still, there has been a long way from continuous EEG to event-related potentials. I think this paragraph now includes the most relevant background work on the development of dog ERP measurement, but it appears rather dismissive on this previous groundwork. As the method is still quite young in dogs, the accumulative information is needed for the field. This section would deserve to be explained in a bit more detail, opening the contribution of these papers for the field and for the present paper. You have succeeded in conducting first scalp-EEG auditory measurements of non-medicated dogs: this is a great achievement, and likely, did not come easily.

Authors’ response:

We are grateful for the evaluation of our revised work. We apologize for leaving out some previous work on dog EEG. We inserted a sentence and references to early and recent continuous dog EEG studies (lines 70-71).

Reviewer 2:

Introduction (ln 45-46): “Studies have recently also shown similarities in the neural correlates of human and dog word processing (12,13).”

-> Ref 13 does not involve neural correlates nor neuroscientific methods, thus should be introduced as a behavioral measure. The authors could also indicate that ref 12 is coming from the same lab. This is both a praise for the group and a sign for the need to replicate the findings on a global level.

Authors’ response:

We changed the sentences according to the suggestion (lines 45-48).

Reviewer 2:

Intro (58-59) “Younger infants (around 14 months) fail to associate phonetically similar words such as bih or dih to different objects in word learning situations (15,16).”

-> As many of the readers do not know the meaning of bih or dih, the authors could indicate the language and English translation of these.

Authors’ response:

We apologize for the misunderstanding. The words bih and dih were nonsense “English” words, i.e. novel words in the referred study. Therefore, they do not have a translation. We added more explanation to the sentence now (lines 60-61).

Reviewer 2:

lines 89-96 and throughout the text: The nomenclature used for the analysis (automatic artifact-rejection: one-step cleaning and a multi-level method for artifact-rejection: three-step cleaning) are non-standard EEG terminology and somewhat misleading in the current form. I suggest calling the simple epoch rejection based on amplitude value not as “automatic”, but as amplitude-based artifact rejection. Also, what is rigorous is not straightforward but open to personal opinions, so I suggest changing the wording of the following sentence accordingly, based on facts:

“According to our knowledge, such a rigorous method has been applied here for the first time on the EEG of awake dogs.”

-> “According to our knowledge, combining visual data exclusion with amplitude-based artifact rejection has been applied here for the first time on the EEG of awake dogs.”

Authors’ response:

We followed the suggestion and changed the naming of the two data-cleaning procedures to amplitude-based and multi-level artifact rejection throughout the main text, in the figures and in the Supplementary material. We also corrected the mentioned sentence (lines 96-98).

Reviewer 2:

lines 120-125: The grand average across conditions in fig s2 is extremely helpful and raises both credibility and reproducibility of the current study, I suggest including it as one of the figures of the manuscript instead of being in supplementary material, where it easily can be lost from most readers. However, it is not advisable to refer to frequency filter as data cleaning procedure here, as it is merely a filter. So, “For better visualization of the general shape of the ERP, we also conducted a third data-cleaning procedure. This was similar to the one-step data cleaning procedure explained before with the exception of the low-pass filter which was set to a lower value (20 Hz).”

→ change e.g. “For better visualization of the general shape of the ERP and easier detection of e.g. the primary auditory responses of the data, we set the low-pass filter to a lower value (20Hz).”

Authors’ response:

In the revised manuscript, we moved Fig. S2 from the Supplementary material to the main text (see Fig. 2), and corrected the sentences as suggested (lines 130-131). However, we would like to make it clear that we first applied a 20 Hz low-pass filter, then trials were rejected using the amplitude-based procedure described for the dataset cleaned with amplitude-based method. This might have not been well explained in the earlier version of the manuscript, therefore, we also included this explanation in the recent version (lines 131-133).

Reviewer 2:

From the fig s2 we can detect the 100 ms primary auditory –like response, which is a big relief. The fact that the 200-ms response appears even larger is puzzling, since the primary responses are usually the most robust of EEG responses. Sometimes the responses appear a

bit oscillatory, though, so this might be that. It would be interesting to see these compared to a pure auditory response, with stimulus such as sine wave. This would confirm whether the 200-300 ms responses have anything to do with speech sounds at all or whether this is some kind of oscillatory echo of the earlier response: these are dogs and we are still at the early days. I suggest adding a little bit more explanation on this part, so that the reader knows the value of this step (confirms that we actually do appear to get a primary auditory-like response of dogs with this method and adds to the reproducibility of the step). This is very important for those who know EEG methodology, for the credibility of the whole field.

In your response letter, you had a reasonable discussion of this issue, I suggest adding it (perhaps slightly modified) also in your manuscript: “In our study, future research could address whether the observed small negativity around 100 ms is similar to an N1 ERP component exhibited by humans. Our experiment was not designed for revealing the N1 component (i.e. we did not expect any modulation of this component), hence, we cannot make any conclusion about it. Future studies could also examine whether the negativity around 200-300 ms is stimulus-specific or also reflects a more general auditory response.”

Authors’ response:

We are grateful for the suggestion. In the revised manuscript, we included the discussion of a possible primary auditory response (lines 135-142).

Reviewer 2:

lines 260-262: “Moreover, even if ERP effects were due to muscle movements, those would still show that dogs differentiate word-categories, hence, such a confound would not undermine the claim about dogs’ ability for discriminating known words from nonsense words.”

-> I agree with this –we don’t need a laborious EEG study to know that dogs differentiate human words, we know they do on the basis of behavior only. However this discussion is important in determining the neural processing: whether we have a window into it with the data or not, HOW is it that dogs process the human words. This is a difference worth mentioning.

Authors’ response:

We corrected the mentioned paragraph by adding that such a confound would not provide electrophysiological evidence for word processing in dogs which was the primary aim of this study (lines 296-299).

Reviewer 2:

Methods, line 457 onward: The authors used a 100ms long sliding window for the statistical analysis, in steps of 50 ms. This is a very long time window in the EEG analysis, since ERP responses can be very quick. In practice, this means that within that time window, the differences calculated can be sums from more than one response (e.g. one positive and subsequent negative potential). This inflates the possible effects of large data drifts, but on

the other hand can help to overcome potential hf noise. It is good to discuss the effects of the analysis choices - there is always a trade-off.

Authors' response:

We used a relatively longer window because ERP effects are usually found in a few hundred milliseconds long time-window in human language studies (e.g. N200-N400 window in Mills et al., 2004). Moreover, one of the studies of human word processing to which we also refer in the manuscript (Friedrich & Friederici, 2005) also used 100 ms windows for data analysis from 400 to 1200 ms after word onset. We added a short discussion about the duration of the time-window and its effects in the Material and Methods section (lines 498-502).

Reviewer 2:

Results&Discussion, ln 118-119 and Methods, ln 478-480 "When significant differences between conditions were found in a time-window, we also tested condition-differences by a non-parametric statistical test (Wilcoxon-signed rank test)." and "Condition differences at selected time-windows were also tested by Wilcoxon signed-rank test. When the repeated-measures ANOVA did not show any interaction effect between channels and condition in a time-window, Wilcoxon signed-rank test was applied on the mean of Fz and Cz. When there was an interaction effect and the post-hoc tests showed an effect only at one of the electrodes, Wilcoxon signed-rank test was calculated for the values of this one electrode."

-> I do not understand the use of Wilcoxon test as a post-hoc for repeated-measured ANOVA here. It is overly liberal for the normal data, Student's t test in its stead is advisable for the normally distributed data.

Authors' response:

The Wilcoxon-tests are not crucial for the analysis of the results because these tests were conducted on condition differences found significant by the repeated-measures ANOVA analysis. The Wilcoxon-tests were originally not included in the manuscript, we included these during the first round of revision. We conducted these tests because Reviewer 2 wrote the following: *"From the current data, it is impossible to determine e.g. if the Grand Average response depicted in Fig. 2A is mainly due to a few dogs' leading effect, or whether the GA indeed represents a valuable average of the whole population."* Our response to this comment was that *"we also conducted Wilcoxon-signed rank tests (see Table S2 in Supplemental Results) in the selected time-windows. This non-parametric test ensures that condition differences are not only due to some extreme values of individual dogs' data."*

We understand that the function of the Wilcoxon-tests might be confusing for the readers. These also do not reveal anything more about our results compared to what is already shown by the ANOVA analysis, therefore, we moved this analysis from the main text to the Supplementary material in the recent version of the manuscript. We also added an explanation in the Supplementary material (Supplemental Results and Discussion). We explained that we test the condition differences found significant by the repeated-measures ANOVA because the non-parametric test might show that condition differences are not only due to extreme values in some of the dog's data. This test operates on the rank of differences between conditions,

hence, the relative magnitude of the condition differences does not influence the results. We also added that on the other hand, this test might be too liberal for normally distributed data.

Reviewer: 1

Comments to the Author(s)

The Editor's letter suggested using bayesian analyses to support the position that ERPs to known and phonologically similar words did not differ. I find this a useful approach, and agree it would be helpful. This is especially relevant as the paper is being showcased for its methodological rigour and approach. Otherwise, I am happy with the revisions.

Authors' response:

We performed Bayesian analysis between the ERP amplitudes of known and phonologically similar words (WORDS and SIMILAR) in those time-windows where we found differences between the ERP amplitudes of the nonsense words and the other two conditions (i.e. between WORDS and NONSENSE and between SIMILAR and NONSENSE). There were two such time-windows (650-750 ms and 650-800 ms) for the data cleaned in three-steps (multi-level artifact-rejection in the recent version of the manuscript), and three such time-windows (200-300 ms, 650-800 ms and 700-800 ms) in the data cleaned in one step (amplitude-based artifact rejection). In all but one time-windows we found substantial evidence for the lack of differences between WORDS and SIMILAR. There was only a weak evidence for no difference in one time-window, between 650-800 ms of the amplitude-based artifact-rejection. We report now the Bayes factor values and their proportional error in the manuscript (lines 185-186; 192-193; 203-206). We added also a short description of the method in Results and Discussion (lines 123-129) and in Materials and Methods (line 520-526). These results support the conclusion of the earlier version of the manuscript. Therefore, we did not change the conclusion in the recent version. We only changed a phrase from “we speculate” to “we propose” in the last paragraph of the Results and Discussion (line 358).